# Hybrid Generative AI for De Novo Design of Co-Crystals with Enhanced Tabletability

**Nina Gubina**[1]    **Andrei Dmitrenko**[1,2]    **Gleb Solovev**[1]

**Lyubov Yamshchikova**[1]    **Oleg Petrov**[1]    **Ivan Lebedev**[3]    **Nikita Serov**[1]

**Grigorii Kirgizov**[1]    **Nikolay Nikitin**[1]    **Vladimir Vinogradov**[1]

[1]ITMO University, St. Petersburg, Russia

[2]D ONE AG, Zurich, Switzerland

[3]Ivanovo State University of Chemistry and Technology, Ivanovo, Russia

`dmitrenko@scamt-itmo.ru`

## Abstract

Co-crystallization is an accessible way to control physicochemical characteristics of organic crystals, which finds many biomedical applications. In this work, we present Generative Method for Co-crystal Design (GEMCODE) [1], a novel pipeline for automated co-crystal screening based on the hybridization of deep generative models and evolutionary optimization for broader exploration of the target chemical space. GEMCODE enables fast *de novo* co-crystal design with target tabletability profiles, which is crucial for the development of pharmaceuticals. With a series of experimental studies highlighting validation and discovery cases, we show that GEMCODE is effective even under realistic computational constraints. Furthermore, we explore the potential of language models in generating co-crystals. Finally, we present numerous previously unknown co-crystals predicted by GEMCODE and discuss its potential in accelerating drug development.

## 1 Introduction

The use of multi-component molecular crystals, specifically co-crystals, have become increasingly popular in various industries including energy [1], electronics [2, 3], optoelectronics [4, 5], food [6], and especially in pharmaceuticals [7–9]. Pharmaceutical co-crystals are defined as solids that are crystalline singlephase materials composed of a drug molecule and an additional pharmaceutically acceptable molecule (coformer) [10]. Co-crystals have a different crystal structure from the original components, leading to unique physicochemical properties. They are appealing because the resulting solid can exhibit better physicochemical properties compared to either of the pure molecules [11]. The formation of co-crystals has been shown to enhance characteristics such as bioavailability [12, 13], solubility [14–16], stability [17–19], pharmacokinetics [20, 21], and mechanical properties [14, 22, 23]. Plasticity is a mechanical property that is particularly important for the pharmaceutical industry. It is known that highly plastic materials tend to produce stronger tablets compared to those exhibiting elastic behavior [24]. In other words, it possesses improved tabletability, defined as the capacity of a powdered material to be transformed into a tablet of specified strength under the effect of compaction pressure [25]. Therefore, it is essential to control for tabletability as it allows direct pressing with minimal addition of excipients to form a stable compact tablet.

Despite all the robustness and versatility of co-crystals, determining the combination of a coformer and parent component with the desired property modification is an extremely non-trivial task, usually

---

[1]`https://github.com/ai-chem/GEMCODE`

addressed by experimental high-throughput screening [26, 27]. Due to the large amounts of time and effort required, such studies remain targeted, focusing on rather narrow classes of candidate compounds.

Artificial intelligence (AI) methods have recently found their way into the field of chemistry [28–32]. Since then, the accumulated experimental data has become the basis for predictive models transforming the traditional way science works. With big data and machine learning (ML), it is now possible to consider a much larger set of candidate molecules for a given problem, rather than being satisfied with a limited number of experiments. Among the pioneering works in the co-crystal domain are the studies aimed at determining the probability of co-crystallization of a particular molecular pair [33, 34]. However, the sole fact of co-crystallization with no information about the properties of the resulting co-crystals is not enough to inform decision making for a specific use case. Accordingly, another direction of research investigated co-crystal properties with AI methods [35, 36]. Still, prediction of most properties has been possible only in the case of already known co-crystallising molecular pairs. *De novo* design of co-crystals with predefined properties leveraging big data to cover a large chemical space remains an actual task of great application value.

Therefore, here for the first time we develop a pipeline that generates coformer candidates based on the structure of a drug molecule to form a co-crystal with predefined mechanical properties. For that, we trained several state-of-the-art generative models on a dataset of 1.75M chemical structures and then fine-tuned them on the state-of-the-art dataset of coformers. We then trained a classical ML model to predict plasticity parameters of the generated coformer candidates. We further employed evolutionary optimization leveraging the trained ML models to improve the tabletability profiles of the generated coformers. Finally, we applied a pretrained graph neural network (GNN) to rank the molecular pairs according to the probability of successful co-crystal formation. We systematically evaluated and optimized the aforementioned individual components to assemble GEMCODE, a practical solution achieving state-of-the-art performance even within computational constraints. The output of GEMCODE is a set of coformers forming a co-crystal with improved tabletability properties for a selected drug compound. Thus, the pipeline can serve as a tool for selecting the best molecular combination of an active pharmaceutical agent and a coformer delivering the desired properties of the co-crystal. In essence, this work makes the following novel contributions:

- We train a transformer-based conditional variational autoencoder (T-CVAE) setting the new state of the art for the coformer generation task, and hybridize it with multi-objective evolutionary algorithm to improve the desired properties of coformers.
- We develop machine learning models for the prediction of mechanical properties of co-crystals for the first time in the field.
- We present GEMCODE, a generative pipeline for *de novo* co-crystal design with target physicochemical properties contributing to drug tabletability.
- In addition, we explore the capabilities of language models in the coformer generation task.
- Finally, we predict a set of molecules forming novel tabletable co-crystals with known drugs.

## 2 Related Work

### 2.1 Generative AI for molecule generation

Traditionally, the process of discovering new molecules or selecting chemical structures for a particular task relies on existing experimental evidence and subjective research experience, both limiting the number and variety of possible compounds to consider. Generative models allow efficient exploration of the molecular space, which has already caused a rapid growth of molecular generative design. Recurrent neural networks [37–40], variational autoencoders [41–44], generative adversarial networks [45–48], evolutionary algorithms [49–53] and hybrid models using reinforcement learning techniques [54–57] have been successfully applied for various problems in chemistry. In this work, we trained, evaluated and compared multiple generation approaches, such as LSTM-based GAN, transformer-based VAE and conditional VAE. The latter was inspired by a study using a conditional VAE model with an attention mechanism to generate molecules [58]. However, our approach differs significantly in that we generated a condition vector based on the predictions of the pretrained gradient-boosting model. In addition, our approach includes a fine-tuning phase on a state-of-the-art dataset of coformers.

DeepMind has recently presented GNoME, an AI tool for generating previously unknown inorganic crystalline materials [59]. Other similar tools exist for inorganic compounds [60, 61]. Our work also lies in the field of solid-state chemistry, but differs in the task of generating coformers, which are small organic molecules. To our knowledge, generative approaches have not yet been applied to produce coformer structures with high co-crystallization potential with drug targets. Our work effectively addresses this problem.

## 2.2 Co-crystal property prediction

Research in co-crystal property prediction is targeted at determining various parameters, such as the lattice energy, density, melting temperature, crystal density, enthalpy and entropy of melting, as well as ideal mole fraction solubility of co-crystals [62–65]. However, a limited number of samples is typically used in the training phase. For example, Gamidi and Rasmuson trained an artificial neural network on the data of 30 co-crystal systems for 8 different drugs [35]. Such models are likely to have very limited generalization power beyond the training data. The most recent model predicting the co-crystal density [36] used a large training set of 4144 molecular pairs covering a much wider chemical space of possible co-crystals. In this work, we predict several mechanical properties of co-crystals for the first time. We use an even larger amount of data for that (6029 samples), which makes our approach more versatile and better generalizable for different pharmaceutical applications.

## 2.3 Applications of language models in chemistry

Large language models have recently been challenged with multiple chemistry tasks, such as property prediction, yield prediction, text-based molecular design, and others [66]. The results suggest that language models are less competitive in generative tasks requiring a deeper understanding of molecular SMILES strings, but show competitive performance in classification and ranking tasks. Another study on the applicability of language models without prior specialization in the chemistry domain found that LLMs can effectively interpret chemical structures given various representations [67]. In addition, the use of language models as agents was explored in ChemCrow [68], which makes chemistry more accessible to researchers with less domain expertise. Following up on these pioneering works, we explore the applicability of language models to the creation of coformer molecules with desired properties, which has not yet been addressed in the past.

# 3 Data

## 3.1 Data collection

**Large dataset of molecules.** In order to train a generative model capable of suggesting reasonable chemical structures, a dataset of molecules from the ChEMBL database (available with CC BY-SA 3.0 license) was collected. From the large variety of molecular structures available in the database, ~1.75M samples were selected using criteria based on the distributions of relevant parameters in the known coformers (Appendix C.1). Using these criteria ensures that the generative models are trained on molecules capable of forming co-crystals.

**Dataset of coformers.** Chemical structures in the ChEMBL database are still substantially different from the structures composing co-crystals. Coformers most often have more basic chemical structures and a smaller variety of functional groups. Therefore, we used an open dataset of 6819 two-component co-crystals [33] (available with MIT license), which contains 4227 unique chemical structures of the coformers, for fine-tuning.

**Dataset of co-crystals mechanical properties.** For the mechanical properties of co-crystals, we used the Cambridge Structural Database (CSD) [69] and a recently proposed protocol for geometric analysis of co-crystalline materials available with a CSD Python API [24]. For each of the 6819 available co-crystals, we used the API to query additional experimental data from the CSD and calculate the following binary parameters of plasticity: presence of non-overlapping Miller planes (Unobstructed planes), presence of orthogonal planes (Orthogonal planes), and presence of hydrogen bonds between the planes (H-bond bridging). Since some of the co-crystals were missing in CSD, this process yielded a total of 6029 records. This data was then used for training ML models to predict each of the three plasticity parameters.

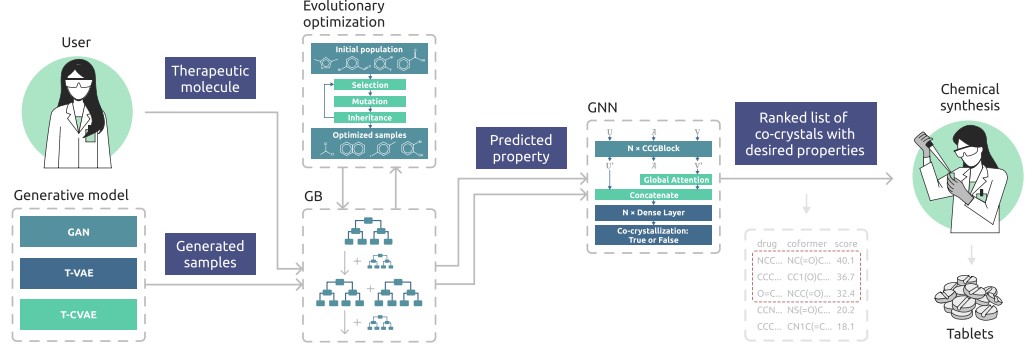

Figure 1: GEMCODE: a pipeline for generative co-crystal design consisting of models (LSTM-based GAN, T-VAE, T-CVAE) generating coformer candidates, gradient boosting (GB) classification models predicting the mechanical properties of co-crystals based on the generated coformers, an evolutionary algorithm producing additional coformer candidates with improved tabletability profiles, and a graph neural network (GNN) ranking co-crystals according to the probability of formation.

We analyzed the number of samples for each plasticity parameter in the collected dataset (Appendix C.2). In the case of orthogonal planes, we observed a dramatic difference between the two groups. When training the corresponding ML model, we accounted for this disproportion by adjusting a threshold probability for predicting a positive class.

## 3.2 Data curation

Cutting-edge generative models use string [70–72], 2D [73–75] and 3D [76–78] molecular graphs as molecular representations. The most common way is the SMILES (Simplified molecular-input line-entry system) notation, as the other approaches have not yet shaped the field to such an extent [79]. Therefore, we used the SMILES representations to describe the composition and structure of chemical molecules with short strings. Additionally, molecular fingerprints allowed us to represent molecules in a vectorized form and compare different structures by calculating a similarity measure (Appendix C.3).

We used RDKit to generate 43 molecular descriptors for each coformer with its SMILES representation. Since co-crystals consist of two coformer components, each one was described by 86 numerical features in total. Before training ML models for the prediction of mechanical properties, we applied a set of preprocessing steps. We engineered new features by aggregating the molecular features of the coformers of the same co-crystal with summation and averaging. To reduce redundancy in the feature space, we investigated the feature importances using embedded methods and the degree of linear association with target variables through correlation coefficients. After feature engineering and filtering, the datasets for the prediction of non-overlapping planes, orthogonal planes, and hydrogen bonding contained 29, 24, and 30 features, respectively.

## 4 GEMCODE: Generative Evolution-based Method for Co-crystal Design

We present GEMCODE, a novel pipeline for generative co-crystal design with improved tabletability properties. It based on the idea of hybridization of deep generative models and combinatorial optimisation. GEMCODE consists of four key components, as depicted on Figure 1.

First, a trained and fine-tuned generative model generates SMILES representations of coformer-like chemical structures. The generated molecules are then fed into the trained ML models along with the therapeutic compounds, where the mechanical properties of co-crystals are predicted. In addition, an evolutionary algorithm is used in combination with the ML models to further improve the tabletability of the generated coformers. Finally, co-crystals with the desired properties are selected for the next step, where a pretrained graph neural network scores and ranks molecular pairs of drugs and coformers according to the probability of co-crystallization. Thus, the pipeline outputs a list of potential coformers with the desired mechanical properties of the co-crystal, ranked according to

the probability of successful co-crystallization. In the following sections, we describe the individual components of the pipeline in more detail.

## 4.1 Prediction of mechanical properties of co-crystals

Since the number of training examples available for prediction of mechanical properties was only 6029, we resorted to the classical machine learning algorithms. We formulated a binary classification problem for each of the mechanical properties and implemented a number of ML models as a first screen, including logistic regression, k-nearest neighbors classifier, support vector machines, decision trees, multilayer perceptron, as well as ensemble models, such as random forest and gradient boosting. We then selected the best models and optimized their hyperparameters to achieve top performance. Those pretrained models were then integrated into the coformer generation and the evolutionary optimization frameworks. To validate this solution, we used an AutoML tool to design the modeling pipeline in an automated way (details are provided in Appendix G.5.

## 4.2 Generation of coformers

The performance of a particular deep neural network is largely determined by its architecture, as well as the strategy to learn the hidden representations [80]. In order to find the most effective solution for the coformer generation task, we implemented and systematically compared three different architectures. Our evaluation included a GAN model with recurrent neural networks for both, generator and discriminator, and two transformer-based models implementing a VAE. For more information regarding the model architectures, refer to Appendix D.4 and D.5.

GAN-based methods consider molecule generation a minimax game, which consists of training a discriminator to distinguish between the real data and the samples produced by a generator (Appendix D.1). In this work, we employed an open-source GAN implementation[2] using LSTM to address molecule generation as a sequence-to-sequence (S2S) problem, inspired by the work of d'Autume [81]. As an alternative, we opted for a transformer architecture [82] as a basis for a VAE, since it normally outperforms recurrent neural network architectures in S2S tasks [83].

Our objective was to produce co-crystals meeting specific tabletability requirements that translate to a set of target mechanical properties. We utilized a conditional variational autoencoder (CVAE) approach [84] to achieve this. By design, CVAE makes it possible to consider physicochemical characteristics of molecules and generate co-crystals with the desired properties (Appendix D.3 offers a more detailed description of the VAE and CVAE models). We used the aforementioned mechanical properties (unobstructed planes, orthogonal planes, and H-bonds bridging) as conditions for CVAE. In the following, we refer to this model as transformer-based CVAE (T-CVAE).

Finally, we included a transformer-based VAE (T-VAE) for comparison, which does not consider any specific properties of molecules, for completeness of the analysis.

## 4.3 Evolutionary optimization of coformers

To increase the quality of coformer generation, we applied a graph-based evolutionary algorithm to structures produced by the generative models. The software implementation is obtained from the self-developed GOLEM library [85]. The fitness function was designed to reinforce the mechanical characteristics of molecules based on predictions of the classification models described above:

$$f(x) = \left(1 - p_u(x), 1 - p_o(x), p_h(x)\right)^T,$$

where $x$ is an evaluated molecule of coformer, $p_u(x)$ is the probability of the positive class for unobstructed planes, $p_o(x)$ is the same probability for orthogonal planes, and $p_h(x)$ – for H-bond bridging. Therefore, minimization of the fitness function $f$ leads to generation of coformer molecules having an improved tabletability profile.

---

[2]https://github.com/urchade/molgen

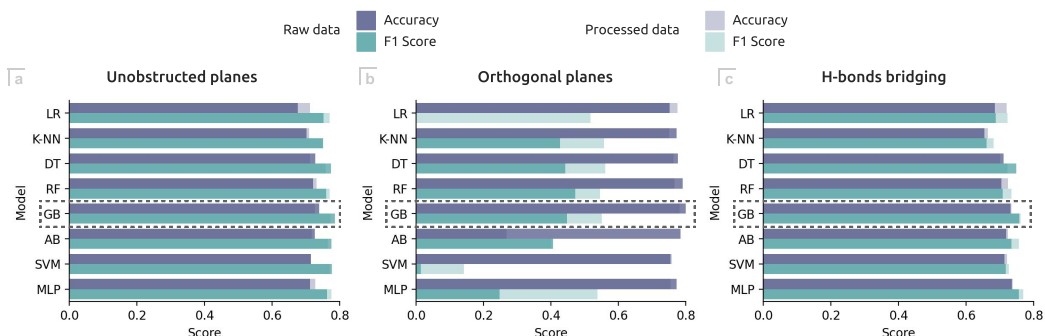

Figure 2: Accuracy and F1 score metrics for the ML models predicting three mechanical properties of co-crystals. (a) Unobstructed planes. (b) Orthogonal planes. (c) H-bonds bridging. The performance of each model is shown before ("Raw data") and after ("Processed data") the feature engineering and feature selection steps.

## 4.4 Estimation of probability of co-crystal formation

Determining the possibility of co-crystallization by molecular pairing is an important step in the co-crystal design. For this reason, many works attempted to solve this problem with AI [86, 34, 87]. Most works that are closely related to our problem do not provide code to reproduce or reuse their results [88–91]. To account for the probability of co-crystallization, we applied an existing GNN-based deep learning framework, called CCGNet [33] (available with MIT license). Unlike many of the previous works, CCGNet achieves state-of-the-art performance predicting co-crystal formation while being 100% open-source and easily reproducible. With an average balanced accuracy of 98.6%, CCGNet efficiently scores and ranks coformer candidates according to the probability of co-crystal formation. Since CCGNet was originally trained on the same database of coformers, we did not perform any fine-tuning and simply integrated the model from the open GitHub repository into the pipeline.

# 5 Experimental studies

## 5.1 Prediction of mechanical properties of co-crystals

**Implementation details.** The preprocessed dataset was randomly split into train and test sets in proportion 4:1. The train set was used to optimize hyperparameters of the models with a grid search using the 10-fold cross-validation (CV). The random grid size was 500 and concerned the following parameters: learning rate, number of estimators, subsample, maximum depth of the individual estimators. The test set was used only once, to evaluate and report the performance of the optimized models. We calculated accuracy and F1 score during the CV to select the best hyperparameter set. The use of the two metrics was important given the imbalanced nature of the "Orthogonal planes" and "Unobstructed planes" target variables (Appendix C.2, Figure 3c). To account for the disproportion, we also adjusted the threshold for the probability of the positive class by calculating precision and recall metrics. Finally, we employed SHapley Additive exPlanations (SHAP) to interpret model predictions, which is based on sensitivity analysis investigating the effect of systematic changes in feature values on the model output [92].

**Results.** Overall, the GB model showed the best accuracy and F1 score compared to the other models across all tasks (Figure 2). Despite the high accuracy for the orthogonal planes parameter, we obtained a moderate F1 score suggesting that the final model is more likely to predict the absence of the orthogonal planes. This is attributed to the disproportion in the training examples discussed earlier. Although we demonstrated a significant improvement in metrics by introducing the probability threshold (Appendix G.2) evaluating the model trained on the processed data, it was not enough to entirely resolve this issue.

We optimized the hyperparameters of the Gradient Boosting (GB) model, which resulted in the performance metrics outlined in Table 10 (Appendix G.4). Furthermore, we conducted a thorough

Table 1: Results of the coformer generation comparison.

| Model | GAN | T-VAE | T-CVAE |
|---|---|---|---|
| Validity (↑), % | 94.57 ± 0.00 | **99.70 ± 0.00** | 98.40 ± 0.00 |
| Novelty (↑), % | 94.90 ± 0.08 | **95.12 ± 0.11** | 80.62 ± 0.25 |
| Duplicates (↓), % | 42.29 ± 0.69 | **24.30 ± 0.45** | 55.70 ± 0.19 |
| Target coformers (↑), % | 2.23 ± 0.17 | 1.68 ± 0.12 | **5.63 ± 0.22** |
| Diversity of target (↑) | **0.30 ± 0.00** | 0.31 ± 0.00 | 0.25 ± 0.00 |

review of the existing research on the prediction of co-crystal properties to compare with our results. Notably, we are the first to develop predictive models for the plasticity parameters, so our metrics set the state of the art. In addition, our work clearly stands out by the number of data points used for training.

With SHAP analysis (Appendix G.3), we learned that the number of atoms among the molecular pairs forming a co-crystal is a decisive factor in the prediction of non-overlapping and orthogonal planes. In both cases, the decrease in the number of atoms in the coformer molecules significantly contributed to the presence of non-overlapping and orthogonal planes. The descriptors associated with the number of hydrogen bond donors (HBD) also had a high degree of importance. As expected, an increase in the number of HBD resulted in the hydrogen bonds forming between planes of the co-crystal.

## 5.2 Generation of coformers

**Implementation details.** The performance of generative models depends on hyperparameters and random restarts [93]. A grid search was implemented to select the best hyperparameters, and multiple trainings were conducted. The generative model was focused on generating coformer-like chemical structures, so it was pretrained on the ChEMBL dataset and then fine-tuned on a dataset of coformers. The importance of fine-tuning was illustrated using t-distributed stochastic neighbor embedding (t-SNE) to visualize the datasets (Appendix D.2). To evaluate the trained models, ten sets of 10,000 molecules were generated and various indicators were calculated, including validity (defined as the percentage of chemically plausible molecules to all generated), novelty (defined as the percentage of newly generated molecules that are not contained within the training set to all generated), percentage of duplicate molecules, percentage of target coformers, and diversity. More details on these indicators can be found in the Appendix F.

**Results.** Analyzing experimental results of coformer generation, we observed that T-VAE produced the highest percent of valid and novel molecules with by far the lowest percent of duplicated structures (Table 1). However, among the generated coformers, only 1.68% had the target tabletability profile, as assessed by the pretrained classification models. In contrast, when generating 10,000 candidates, T-CVAE produced 5.63% of new coformers with the required mechanical properties on average. While the diversity of target coformers [3] was slightly higher for GAN, it was able to produce the intermediate 2.23% of such coformers. Therefore, we conclude that T-CVAE was the most effective approach to target coformer generation. However, the transformer architecture was also the most demanding for both, the training and the generation phases (see Appendix D.6 for more details).

Ultimately, we recommend to use an ensemble of generative models whenever sufficient computational resources are available. Our findings presented in Appendix D.7 suggest that the three models produce complementary results. Collectively, GAN, T-VAE and T-CVAE generate up to 2.47 times more unique target coformers than individually.

**Additional experiments with language models.** Inspired by the most recent applications of language models in chemistry [66–68, 94] we investigated their potential in the coformer generation task. First, we employed a reduced GPT-2 model with eight heads, four attention blocks, and 14.7M parameters. Similarly to other models, GPT-2 was pre-trained on the ChEMBL dataset and then

---

[3]A numerical comparison of the target molecules generated by the models and those available in the training dataset is given in the Appendix D.7

Table 2: Results and statistical significance of the evolutionary optimization.

| Model | Property | Median probability (↑) | | $p_{adj}$ | Novelty (↑) |
|---|---|---|---|---|---|
| | | Generated | Optimized | | |
| GAN | Unobstructed planes | 0.82 | 0.82 (+0.0%) | - | |
| | Orthogonal planes | 0.37 | **0.39 (+5.4%)** | 2.68e-11 | 0.68 |
| | H-bond bridging | 0.62 | **0.69 (+11.3%)** | 1.05e-66 | |
| T-VAE | Unobstructed planes | 0.82 | 0.82 (+0.0%) | - | |
| | Orthogonal planes | 0.38 | 0.40 (+5.3%) | 2.71e-9 | **0.72** |
| | H-bond bridging | 0.64 | 0.69 (+7.8%) | 1.76e-65 | |
| T-CVAE | Unobstructed planes | 0.82 | **0.83 (+1.2%)** | 9.52e-05 | |
| | Orthogonal planes | 0.38 | 0.39 (+2.6%) | 1.88e-9 | 0.60 |
| | H-bond bridging | 0.64 | 0.69 (+7.8%) | 1.82e-46 | |

fine-tuned on the coformers dataset (see Appendix D.9 for more details). We observed that GPT-2 produced significantly lower percent of new and valid structures per 10,000 generations (Appendix D.10). Nevertheless, the model achieved 3.32% of new molecules with the desired physicochemical properties, which is comparable to GAN and T-VAE. These results prompted us to further test a more recent and capable language model. Therefore, we trained Llama-3-8B with the low-rank adoption (LoRA) algorithm using the same ChEMBL and coformer datasets. We observed major improvements in validity, novelty and the number of duplicates among the generated molecules compared to GPT-2. Notably, Llama-3-8B produced the maximum diversity percent compared to all the tested generative approaches. However, the number of molecules with target physicochemical properties dropped to only 0.34%. Analyzing these empirical results, we conclude that language models show good potential in the coformer generation task but have to be heavily optimized to achieve competitive performance with GEMCODE. We leave this endeavour for the future work.

## 5.3 Evolutionary optimization of coformers

**Implementation details.** The multi-objective optimization algorithm used in this work considers molecules as undirected graphs and follows the generational evolutionary scheme MOEA/D [95]. First, a population of individuals is evaluated with the fitness function. Then, MOEA/D-based selection is applied to pick individuals from the population to undergo mutation. After the variation by mutation is done, the inheritance operator is used to form the new population of individuals to proceed to the next iteration (see Appendix E.1, E.2 for more details).

To choose an effective evolutionary scheme for the task we compared SPEA-2 [96] and MOEA/D (see Appendix E.5). Experiments have shown MOEA/D obtaining better results in some cases.

The initial population of coformer structures (obtained with the previously described generative models) were varied by the set of mutation operators, inspired by the work of Leguy [50]. The set of mutations includes simple operations (add, delete, or replace an atom, delete or replace a bond) and more complicated, multi-step actions (delete or move a functional group, insert carbon, remove an atom if it has only two neighbors). See Appendix E.3 for more details on optimization runs.

**Results.** To evaluate results of the evolutionary search, we compared the probabilities of coformers to possess the desired mechanical properties before ("Generated") and after ("Optimized") evolutionary optimization (Table 2). For that, we used the pretrained ML models to retrieve the probabilities, calculated statistics and applied the non-parametric one-sided Mann-Whitney test (see Appendix E.4, F.1 for more details). In most cases, we observed a significant increase in the median probability[4] of the target class. Notably, evolutionary optimization equalized the performance of different generative models in their ability to produce coformers with the target tabletability profile. Moreover, this process consistently yielded new coformer structures, not present in the training set or in the initial population.

---

[4] Median probability score can be best described as the median probability of assigning coformers to a positive class for each of the mechanical properties. In other words, median probability score gives an idea about the central tendency of the model's confidence in predicting a particular mechanical property.

Table 3: Experimentally validated coformers improving drug tabletability generated by GEMCODE. SMILES were selected based on two tabletability parameters (Unobstructed planes, H-bond bridging) and similarity metric (IT = 1).

| Drug | Generated SMILES | CSD Refcode | Model | Ref. |
|------|------------------|-------------|-------|------|
| Nicorandil | O=C(O)C=CC(=O)O | WAHGEV | GAN / T-VAE / T-CVAE | [97] |
| Rivaroxaban | O=C([O-])CC(=O)[O-] | YORVEJ | T-VAE | [98] |
| Paracetamol | C1=CC=C2C=CC=CC2=C1 C[N+](C)(C)CC(=O)[O-] | LUJSIT CUQKAC | GAN / T-VAE / T-CVAE T-CVAE | [99] [100] |

## 5.4 Validation case studies

In order to test the effectiveness of GEMCODE, we generated coformers for the drugs with poor ability to form a tablet by powder pressing. Among the therapeutic molecules selected for the pipeline validation were Nicorandil, Rivaroxaban and Paracetamol. For each of the listed drugs, experimentally validated molecules were found among the GEMCODE-generated coformers improving tabletability of the co-crystals (Table 3). More details can be found in Appendix B.1.

## 5.5 Novel coformer molecules predicted by GEMCODE

To showcase the ability of GEMCODE to predict novel coformers with target tabletability profiles, we generated coformers for one of the therapeutic molecules, i.e., Nicorandil. GEMCODE enabled discovery of 23 unique coformer with improved mechanical properties and with the presence of functional groups as in experimentally validated tabletable co-crystals (see Table 4 in the Appendix B.2). This result demonstrates the potential of GEMCODE as an indispensable tool for accelerated drug development. Broader impact is further discussed in Appendix A.

## 6 Limitations

The evidence presented above looks very promising for the practical applications of our pipeline. However, a comprehensive experimental validation involving organic synthesis of coformers and co-crystal formation followed by a tablet compression experiment is required to confirm its utility. Based on our empirical results, we anticipate the following limitations of the proposed pipeline:

- The coformers molecular space may be too narrow for some applications due to the small sample size of the coformer dataset. Nevertheless, if computational power is available, it is possible to use an ensemble of generative models, which partially solves the problem by increasing the number of unique molecules generated.

- Currently, the GB model is biased towards predicting the absence of orthogonal planes, leading to more false negatives in the predicted coformers. We recommend exploring an alternative set of coformers based on the other two mechanical properties only.

- Low-scale screening may still result in some coformers failing to form co-crystals, particularly those optimized through evolution. Screening more coformers increases the chances of finding co-crystal pairs for a specific therapeutic agent.

- While polymorphism's impact on predicting co-crystal mechanical properties is not examined here, its significance is undeniable and often understated. Despite limited reported polymorphs, their potential impact on prediction model accuracy in the co-crystal field necessitates further exploration, considering the current scarcity of polymorphism data.

Most limitations of the proposed pipeline can be solved with more data available for training, which remains a major challenge for successful AI applications in co-crystallization. We are working towards collecting more data and improving its quality. Also, to date GEMCODE has been adapted mainly for pharmaceutical applications. In the future, we plan to extend GEMCODE by adding more predicted physicochemical properties and other crystal forms to be able to expand beyond the pharmaceutical field.

# 7 Conclusion

In this work, we presented GEMCODE, a novel generative pipeline for *de novo* co-crystal design. To make it as effective as possible, we implemented the hybrid generative approach combininig positive sides of both deep learning models and combinatorial optimisation. We systematically evaluated and discussed the individual components of the pipeline achieving state-of-the-art performance in the corresponding tasks. Furthermore, we performed experiments to validate the pipeline by generating coformers for three different drugs and discovering previously unknown coformers for Nicorandil. In addition, we explored the applicability of language models in the coformer generation task and identified prospective research directions. Despite limitations associated with data availability, GEMCODE enables fast generation of unique and valid chemical structures of coformers with high probabilities of co-crystallization and target tabletability profiles. This research enhances co-crystal design for pharmaceuticals and contributes to the accelerated drug development. Thanks to data and code availability, our versatile hybrid approach might find other impactful applications in chemistry.

# 8 Acknowledgements

This research is financially supported by the Foundation for National Technology Initiative's Projects Support as a part of the roadmap implementation for the development of the high-tech field of Artificial Intelligence for the period up to 2030 (agreement 70-2021-00187)

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

# Appendix

## A Impact statement

This paper presents a new method for the generative design of organic co-crystals with the goals to advance application of machine learning to pharmaceutical co-crystal design and to accelerate and reduce cost of development of solid forms of active therapeutic molecules. Extensive experimental results and multiple case studies described in the paper provide hard evidence of the effectiveness of our approach. Therefore, we are confident that this work can have a broader impact on drug discovery and development, pharmaceutical industry in general and other related domains.

While we identify the aforementioned societal impacts as strongly positive, there is a risk of malicious and unintended use, as well as inaccurate predictions affecting decision-making in the drug manufacturing process. However, we deem the potential negative impacts limited due to the complexity of regulations in the corresponding fields and the laboratory experiment being the ultimate measure of success.

## B Results

### B.1 Validation experiment

The validation experiment involved generation of 10000 coformers using ensemble generative models, namely GAN, T-VAE, T-CVAE. The generated candidates in combination with one of the three drugs (Nicorandil, Rivaroxaban, Paracetamol) were labeled into classes of three mechanical plasticity parameters. The criterion for getting into the final dataset was satisfaction of Unobstructed planes and H-bond bridging parameters. Experimentally validated coformers among the generated molecules were then searched for using the Index Tanimoto (IT), which is a metric of molecular structure similarity. Coformers with IT = 1 were compared with molecules from literature data and added to the Table 3.

**Nicorandil.**   Nicorandil, a medication that dilates blood vessels, is prescribed for treating angina pectoris, a condition characterized by chest pain caused by temporary reduced blood flow to the heart muscle. Unfortunately, during the manufacturing of Nicorandil tablets, the drug can degrade chemically due to the heat produced at high compressive pressure. Generation of a set of coformers for this drug resulted in a fumaric acid molecule among them. Experimental findings have demonstrated that co-crystallizing Nicorandil with fumaric acid not only led to the success of co-crystallization but also improved tabletability properties [97].

**Rivaroxaban.**   Rivaroxaban is an anticoagulant medication that is used to prevent blood clots Rivaroxaban is often taken orally by patients, but the drug is poorly suited for direct compression tableting. The generation of coformers for rivaroxaban using GEMCODE led to the detection of malonic acid among the resulting molecules. According to Kale et al. the formation of this co-crystal leads to its excellent plastic deformation under applied compaction pressure, resulting in successful tablet formation [98].

**Paracetamol.**   Paracetamol is an analgesic and antipyretic from the group of anilides, but forms an unstable tablet by direct pessation. Among the coformers generated by GEMCODE was found experimentally confirmed coformers of paracetamol are naphthalene and trimethylglycine (betaine) [99, 100]. They are not only able to form a co-crystal with paracetamol, but also improves its tabletability, as demonstrated in Karki et al. and Maeno et al..

### B.2 Discovery experiment

For the therapeutic molecule Nicorandil, the labeled generated candidates were selected by meeting the conditions of being recognized as safe for human use and presence of carboxyl functional group, which means finding molecules forming the same synthon as experimentally confirmed coformers. The discovered coformers are presented in the Table 4, which is divided into blocks depending on each model used for generation. The CCGNet score column indicates the values of the ranking results

Table 4: Previously unknown novel coformers generated using GEMCODE to improve the tabletability of the drug Nicorandil. SMILES are selected based on a similarity metric (IT $\geq$ 0.7). Target properties abbreviated as follows: Unobstructed planes (U), Orthogonal planes (O), H-bond bridging (H).

| Model | Generated SMILES | Target properties | CCGNet score |
|---|---|---|---|
| GAN | CC1=CC=C(C(=O)O)C=C1 | U / H | 46.70 |
| | CC(=O)NC1=CC=CC(C(=O)O)=C1 | U / H | 44.94 |
| | COC1=CC=CC=C1C(=O)O | U / H | 44.84 |
| | O=C(O)C1=CC=CC=C1 | U / H | 44.45 |
| | O=C(O)C1=CC=CN=C1 | U / H | 42.67 |
| | O=C(O)C(=O)O | U / O / H | 42.51 |
| | COC1=CC=C(C(=O)O)C=C1 | U / H | 40.64 |
| | COC1=CC=CC(C(=O)O)=C1 | U / H | 39.45 |
| | O=C(O)COC1=CC=CC=C1 | U / H | 35.36 |
| | CC(=O)NC1=CC=C(C(=O)O)C=C1 | U / H | 33.34 |
| | O=C(O)CO | U / H | 28.60 |
| | O=C(O)CC1CCCCC1 | U / H | 24.29 |
| | O=C(O)C1=CC=NC=C1 | U / H | 23.40 |
| | O=C(O)CC(=O)O | U / O / H | 21.99 |
| | O=C(O)CC1=CC=CC=C1 | U / H | 21.94 |
| VAE | O=C(O)C1CCCCC1 | U / H | 59.53 |
| | O=C(O)C1=CC=NC=C1 | U / H | 51.93 |
| | COC1=CC=C(C(=O)O)C=C1 | U / H | 51.32 |
| | COC1=CC=CC=C1C(=O)O | U / H | 49.99 |
| | O=C(O)CC1=CC=CC=C1 | U / H | 48.50 |
| | O=C(NC1=CC=CC=C1C(=O)O)C1=CC=CC=C1 | U / H | 48.27 |
| | O=C(O)COC1=CC=CC=C1 | U / H | 47.53 |
| | O=C(O)C(=O)O | U / O / H | 46.70 |
| | CC1=CC=C(C(=O)O)C=C1 | U / H | 44.86 |
| | O=C(O)C1=CC=CC=C1 | U / H | 44.45 |
| | CC(=O)NC1=CC=C(C(=O)O)C=C1 | U / H | 42.07 |
| | O=C(O)C=CC1=CC=CC=C1 | U / H | 40.01 |
| | O=C(O)C=CC1=CC=C(O)C=C1 | U / H | 37.96 |
| | O=C(O)CC1CCCCC1 | U / H | 32.38 |
| | O=C(O)C1=CC=CN=C1 | U / H | 25.17 |
| | CC(O)C(=O)O | U / H | 24.27 |
| | O=C(O)CO | U / H | 22.18 |
| | O=C(O)CC(=O)O | U / O / H | 22.01 |
| CVAE | COC1=CC=C(C(=O)O)C=C1 | U / H | 46.34 |
| | O=C(O)C(=O)O | U / O / H | 42.97 |
| | O=C(O)C1=CC=CC=C1 | U / H | 35.53 |
| | O=C(O)CC(=O)O | U / O / H | 33.68 |
| | CC1=CC=C(C(=O)O)C=C1 | U / H | 32.85 |
| | O=C(O)CC1=CC=CC=C1 | U / H | 31.68 |
| | O=C(O)CNC(=O)C1=CC=CC=C1 | U / H | 31.41 |
| | O=C(O)CCC(=O)C(=O)O | U / H | 28.01 |
| | O=C(O)C1CCCCC1 | U / H | 24.09 |

for the probability of co-crystallization with Nicorandil for the candidate molecules. The ranking was performed within the set of molecules generated by each model separately.

**Coformer analysis.** Besides carboxyl, the demonstrated chemical compounds contain functional groups such as hydroxyl and amide groups, which are characteristic of the confirmed coformers of Nicorandil from the work of Maeno et al. The generated coformers contain various structural modifications, such as changes in the length of the carbon skeleton, addition and partial substitution of functional groups, the appearance of multiple bonds and benzene rings. It is important to understand that GEMCODE is focused on the search for the best candidates from the point of view of crystallography and does not address the deep issues of interaction of the created structures with the human body. Therefore, with the help of expert evaluation, we selected molecules that should normally be safe for use in pharmaceuticals.

Meanwhile, many of the discovered coformer structures not only already meet all three mechanical parameters consistent with improved tabletability properties, but also have a high probability of successful co-crystalization with Nicorandil. In this way, the process of discovering new coformers using GEMCODE can be described as a smart approach to selecting candidate molecules for property-controlled co-crystallization.

## C  Data

### C.1  Molecule selection criteria

The ChEMBL database contains information on more than 2.4 million drug-like chemical compounds. For training generative models, we needed a large number of molecular structures that would have similar properties to the known coformers. Therefore, 1.75 million samples were selected from the molecular structures of the ChEMBL database according to the following criteria:

- Structural type: molecule.
- Class: small molecules.
- Molecular weight of each component <600 Da.
- Number of hydrogen bond donors (HBD) less than 3 and hydrogen bond acceptors (HBA) less than 8.
- Number of rotatable bonds up to 9.
- Polar surface area up to 138 nm.
- Number of heavy atoms in molecular structure up to 39.

### C.2  Co-crystal data

Mechanical properties of the co-crystals determine their viscoelastic nature. The presence of unobstructed planes and additional slip planes orthogonal to the stacked layers lead to the improved plasticity [101]. Also, there exists evidence that the lack of hydrogen bonding between the layers has a positive effect on the plasticity of a crystal [102, 103]. Therefore, an "ideal" co-crystal in terms of plasticity should have non-overlapping slip planes, additional orthogonal planes and no hydrogen bonding between the planes (Figure 3a).

The compaction properties of many pharmaceutical powders depend on their viscoelastic nature. The closer the material to a perfectly plastic body, the larger the bonding area after compaction of the powder and the denser (less porous) the compressed tablet is (Figure 3b). Therefore, accurate prediction of the plasticity parameters is essential for data-driven co-crystal design.

### C.3  Representation of molecules

Traditionally, molecules are represented as structural diagrams with bonds and atoms, but such representations are not well suited for efficient computation. Alternatively, molecules can be represented with SMILES and molecular fingerprints, which have been extensively used for various applications, including the generative models [104]. SMILES notation is often used to describe the composition

and structure of a chemical molecule by means of short strings (Figure 4a). Whereas molecular fingerprints is a way of representing molecules in the vectorized form (Figure 4b). Therefore, molecular fingerprints enable comparing different structures by calculating similarity measures.

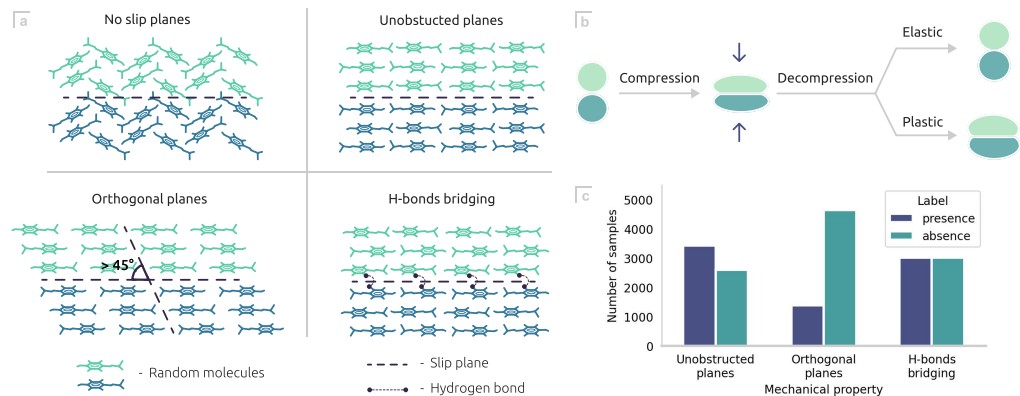

Figure 3: (a) Schematic representation of the mechanical properties of co-crystals. No slip plane and H-bond bridging are associated with low tabletability. The other two properties positively correlate with tabletability. (b) Schematic representation of the particle deformation during powder compression. (c) Number of coformer samples of each category per mechanical property.

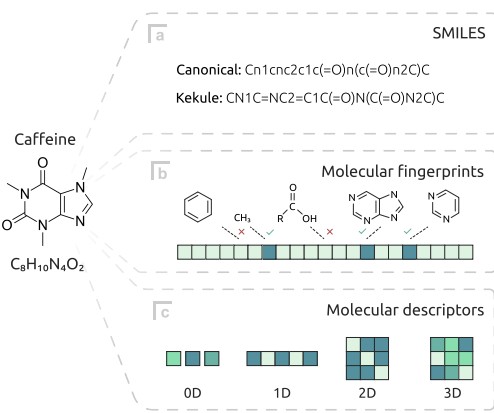

Figure 4: Molecular representation using the chemical structure of caffeine as an example in the form of SMILES, molecular fingerprints, and molecular descriptors.

# D    Generative models

## D.1    GAN

GANs typically consist of two neural networks, a generator and a discriminator, playing an adversarial game against each other while learning the data distribution $p^*(x)$. The generator network receives a random input signal and generates data distribution $p_\theta(x)$, while the discriminator network $D_\phi(x)$ evaluates the generated data and tries to distinguish it from the real training examples [105]. In the original formulation, both networks are improved by competing with each other following a min-max optimisation procedure:

$$\min_\theta \max_\phi E_{p^*(x)}[log D_\phi(x)] + E_{p_\theta(x)}[log(1 - D_\phi(x))].$$

Goodfellow et al. proposed alternate generator losses providing better gradients for the generator [106]:

$$E_{p_\theta(x)}[-log(D_\phi(x))].$$

Since 2014, GANs have been successfully used for numerous applications, including modeling of astronomical phenomena [107], experiments in particle and high-energy physics [108], medical imaging [109], and molecule generation [45, 46]. The GAN takes SMILES representations of molecular structures as input. In the training process, the generator network creates molecular representations from the Gaussian noise and the discriminator network tries to differentiate those from the tokenized SMILES of the real chemical compounds. As a result, the generator learns to output new molecular structures similar to those in the training set.

## D.2   GAN training and fine-tuning

The GAN trained on the ChEMBL dataset with batch size of 512 and learning rate of 0.001 consistently produced molecules with validity >75 % after 25,000 training steps. After 30,000 steps, this model was fine-tuned on the coformer dataset with a smaller batch size of 256 for additional 1,000 steps (Figure 5a). The t-SNE analysis reveals that the molecular space of coformers is considerably more constrained compared to that of ChEMBL (Figure 5b). Therefore, fine-tuning was critical to shift chemical compound generation towards the molecular space of coformers. The final model was able to produce >95% of valid and >86% of unique chemical structures molecules in the test generation of 1000 molecules at 5 times repetition.

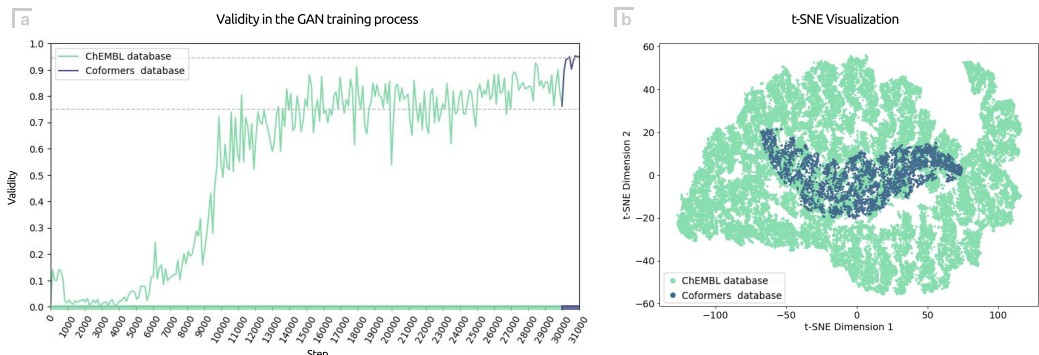

Figure 5: GAN training results on ChEMBL datasets and coformers: (a) plot of the growth of the valid chemical structures share in a batch, (b) t-SNE visualization of molecules from the ChEMBL dataset and coformers.

## D.3   VAE and CVAE

Variational Autoencoders (VAEs) consist of two deep neural networks, namely, an encoder and a decoder. The encoder network takes an input feature vector and converts it into a fixed-dimensional vector, while the decoder network converts this fixed-dimensional vector back to the original input feature vector. The primary objective of an autoencoder is to learn an identity function, and the fixed-dimensional vector is referred to as the latent vector $z$. This latent vector $z$ serves as an information bottleneck, meaning that it is designed to capture only the most statistically salient information in the data. In VAEs, the latent vectors $z$ are sampled from a normal distribution $N(0, I)$, where I is the identity matrix. To train VAEs, the loss function, which needs to be optimized for input data $X$ and latent vector $z$, can be formulated as follows:

$$E[logP_\theta(X|z)] - D_{KL}[Q_{\theta'}(z|X)||N(z)],$$

where $D_{KL}$ is the Kullback–Leibler divergence, which measures the difference between two probability distributions, $Q$ and $N$; $E$ is the mathematical expectation; $P$ and $Q$ are probability distributions.

The probability distributions $P_\theta(X|z, c)$ and $Q_{\theta'}(z|X)$ are learned by deep neural networks called the decoder and encoder, respectively. These networks have learnable parameters $\theta$ and $\theta'$. The first term of the loss function is the reconstruction error for the input data $X$. In contrast, the second term measures the similarity between the probability distribution of the latent space and the target probability distribution, $N(z)$, which is $N(0, I)$.

When using VAEs, it is difficult to control the specific properties of the generated data. Also, since the latent vector is sampled from a unimodal Gaussian distribution, the generated objects tend to be very similar to the training examples. This is not an efficient way to generate new molecules.

Conditional variational autoencoders (CVAEs) were developed to address these challenges. CVAEs can learn multimodal probability distributions by adding a condition vector as an additional input during the generation process. The objective function of a CVAE with condition vector $c$ (passed as an input to the encoder and the decoder) is given by:

$$E[logP_\theta(X|z, c)] - D_{KL}[Q_{\theta'}(z|X, c)||N(z)].$$

### D.4 VAE and CVAE with Attention

It is common knowledge that generating long sequences can be a challenging task for recurrent neural networks. Therefore, we considered transformers, as a more modern and effective architecture for the task. In our approach, we apply the Pre-Layer Normalization Transformer [110], a modification of the original Post-Layer Normalisation Transformer. Similarly to VAE, it is composed of two neural networks, an encoder and a decoder, but with the attention mechanism. Such architectures are known to suffer from a posterior collapse [111]. To overcome this, we used Kullback-Leibler divergence annealing (KLA) [112]. Ultimately, the loss function of the T-CVAE is given by:

$$E[logG_{\theta'}(X_t|z, X_{dec}, c)] - k_w D_{KL}(Q_\theta(z|X_{enc}, c)||p(z|c)),$$

where $D_{KL}$ is the KL divergence; $E$ is the mathematical expectation; $Q_\theta$ is a parameterized encoder function; $Q_{\theta'}$ is a parameterized decoder function (generator); $p(z|c)$ is a conditional Gaussian prior. Here, $\theta$ , $\theta'$ , $X_{enc}$ , $X_{dec}$ , $z$ , $X_t$ , $c$ , $k_w$ are the parameter set of the encoder, the parameter set of the decoder, the input of the encoder, the input of the decoder, the latent variables, the reconstruction target, the conditions, and the weight for KLA, respectively. This objective function was inspired by the work of Kim [58].

### D.5 Proposed architectures of T-VAE and T-CVAE

Our proposed architectures of T-VAE and T-CVAE are shown in Figure 6. Since T-CVAE is an upgrade of T-VAE, the schematics are virtually identical. They differ only by the presence of a block responsible for concatenating an additional condition vector with physicochemical properties with the latent space vector in yellow and the molecule token vector at the input to the encoder. Thus, the architecture represents a language model of a transformer whose encoder encodes information about molecules in a 128-dimensional conditional Gaussian latent space. In case of T-CVAE, a vector of conditions consisting of physicochemical properties predicted by gradient boosting model is attached to the latent space. Based on this vector, the transformer decoder learns to generate coformer candidates. The molecule tokens are embedded into 512 dimensions, the same as the model dimension. The encoder and decoder in the proposed transformer architecture consist of 6 layers of 8 heads each.

### D.6 Comparison of computational costs

Table 5 compares steady GPU memory consumption while training, training time (10 epochs for GAN and 30 epochs for T-VAE/T-CVAE), as well as the time required to generate a single molecule. It is noteworthy that the Beam Search method with beam size $b = 4$ was used for molecule generation in T-VAE/T-CVAE, which significantly increases the generation time.

To put this into perspective of our study, T-CVAE required about 45 minutes on an NVIDIA RTX A6000 graphics card to generate 10,000 molecules. GAN managed to do the same in 1.88 seconds. A similar evaluation on a more common NVIDIA GeForce RTX 2070 resulted in 3.73 hours and 3.37 seconds, respectively. Arguably, this makes T-CVAE practically infeasible for many users. Taking

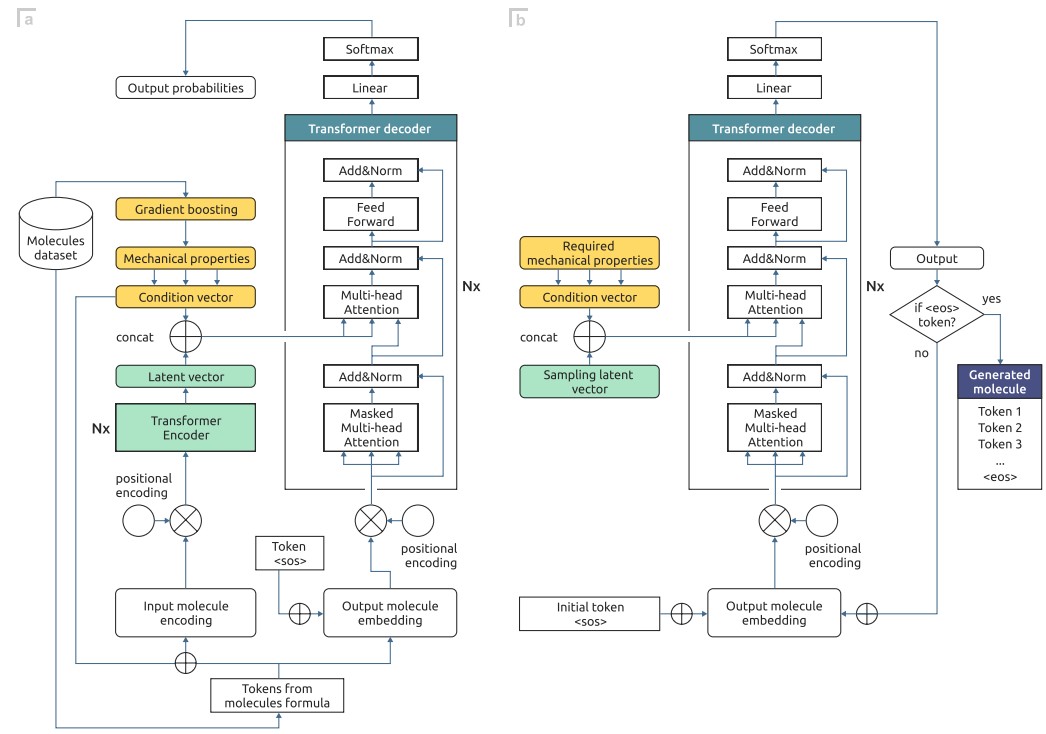

Figure 6: The architecture of T-VAE/T-CVAE for (a) train and (b) generation pipeline.

Table 5: Comparison of GPU memory usage, training and generation times.

|  | GAN | T-VAE | T-CVAE |
|---|---|---|---|
| GPU memory (GB) | **6.40** | 8.00 | 8.10 |
| Training time (hours) | **2.82** | 22.66 | 22.68 |
| Generation time (ms/molecule) | **0.19** | 270.00 | 270.00 |

this into account, we recommend a pragmatic choice to keep GAN as the default generative model in GEMCODE. However, a combination of generative models is required to achieve broader exploration of the target chemical space.

### D.7 Additional result of comparing models

Testing whether the generative models produce the same molecules was a fascinating experiment. In our study, we performed ten generations of 10,000 molecules each, 100k molecules for each model in total. Then, we filtered molecules by discarding duplicates, chemically invalid molecules, not new molecules, and molecules that do not satisfy the required physicochemical properties and SA. After filtering, we got the following result: out of 100k molecules, GAN generated 1639, T-CVAE – 2452, and T-VAE – 1407 molecules. Among all of those, only 202 molecules were found to be common. A more detailed evaluation of the intersection of the generated molecules can be seen in the Figure 7. Thus, in this experiment, each model was able to generate unique molecules that the other models did not produce. Specifically, GAN generated 1051, T-CVAE – 1733, and T-VAE – 698, making the total of 3482 new unique coformer molecules with the properties of interest. Therefore, taken together, the models can generate 1.42 times more molecules than T-CVAE, 2.47 times more than T-VAE, and 2.21 times more than GAN. According to these empirical results, if necessary and under certain research conditions, it may be relevant to use an ensemble of generative models to increase the total number of generations of diverse molecules. Of course, this approach is inherently resource-intensive.

Table 1 shows the values of target molecules in percent. These percentages can be interpreted as the probability with which a generative model can create a new molecule with the properties of interest in a single generation iteration.

In order to avoid misunderstandings, we also present below a numerical comparison with the target molecules in the training dataset.

To interpret the efficiency of generative models, we present a quantitative comparison of target molecules in the training dataset. A dataset with examples of existing pairs of co-crystals, consisting of 4200 molecules, was used for additional training for the task of generating co-crystals. This dataset contains 355 molecules, which correspond to our selection conditions for the formation of a co-crystal with Theophylline. As can be seen from the description above and Figure 7, even the least efficient model was able to synthesize 1639 new potential molecules during the experiment. Thus, it can be seen that the use of the generative models developed by us to search for a coformer is very promising.

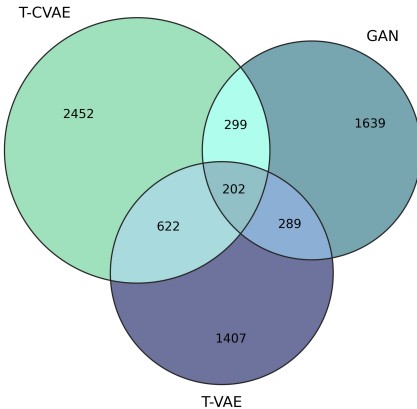

Figure 7: How unique molecules created in different models intersect.

Also, it is worth noting that if we repeat the experiment and generate another 100,000 molecules, we will be able to obtain a significant number of more potential molecules. Since we have not conducted this additional study, and we cannot determine exactly what the limit of each model in generating unique molecules is, we cannot calculate any relative metrics to compare model performance with respect to the training data. For this reason, we propose numerical values.

## D.8 Similarity analysis of the generated molecules

In order to further analyze the novelty of the generated molecules, we performed additional experiments to calculate the Tanimoto similarities for the generated molecules. For this purpose, we plotted histograms illustrating the distribution of maximum IT between the generated coformers and the coformers from the training dataset (Figure 8a). Notably, the distribution is mostly centered on IT values between 0.5 and 0.61 for all generative models. This observation strongly supports the claim that the generated molecules exhibit substantial novelty. The Tanimoto similarity distributions between all molecules generated by GAN, VAE, and CVAE were also analyzed (Figure 8b). For each model, the average Tanimoto similarity ranges from 0.70 to 0.75. On the one hand, this indicates a sufficient diversity of molecules. On the other hand, the relatively high average similarity was expected, since all generated coformers refer to the formation of a co-crystal with the same drug. This fact agrees well with the observation that the CVAE distribution is skewed towards 1 due to the "condition" architecture block that enhances the drug-specific targeting properties of the coformers.

## D.9 Details on training language models

GPT-2 was pretrained on the ChEMBL dataset with a batch size of 128 for five epochs and then fine-tuned on the coformers dataset with a batch size of 256 for 15 epochs. We used the top-p (nucleus) sampling method with $p = 0.95$, as recommended in the study by Holtzman [113] for balanced quality, diversity, and coherence in the model outputs.

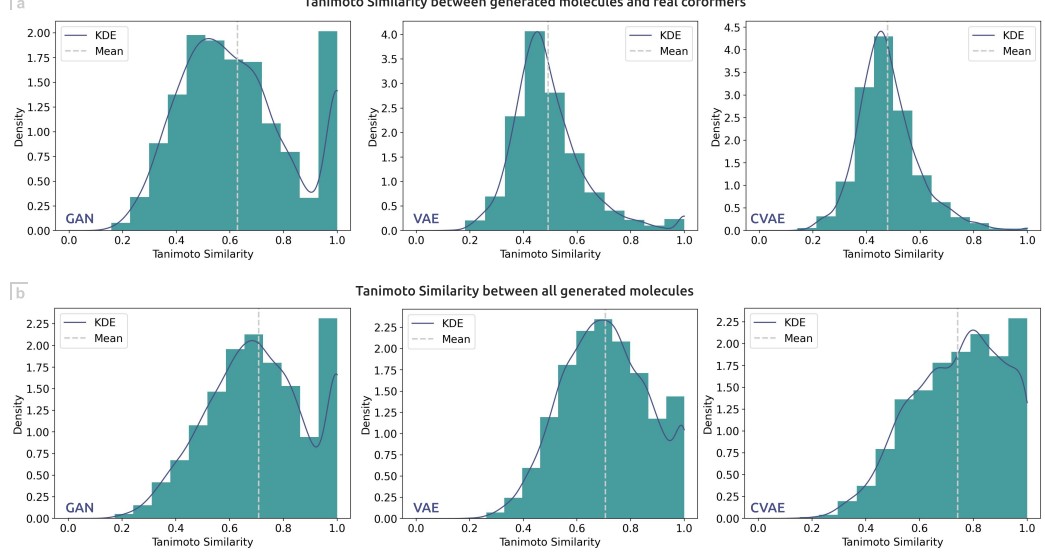

Figure 8: Tanimoto Similarity Histograms: (a) for generated molecules and real coformers, (b) for all generated molecules.

We trained Llama-3-8B with LoRA [114] and 4-bit quantization. It was pretrained on the ChEMBL dataset for 250 epochs and then fine-tuned on the coformers dataset for 100 epochs. Standard Llama-3 tokenizer was used, which likely was the main reason for moderate performance. The total LoRA training time was about 20 minutes.

### D.10 Evaluating and comparing language models for coformer generation

We evaluated generations of GPT-2 and Llama-3-8B models and compared them to the best scores produced by the generative models in GEMCODE (Table 6).

Table 6: Comparison of coformer generation: GEMCODE vs. language models.

| Model | GEMCODE | GPT-2 | Llama-3-8B |
|---|---|---|---|
| Validity, % | **99.70 ± 0.00** | 92.30 ± 0.20 | 98.3 ± 0.00 |
| Novelty, % | **95.12 ± 0.11** | 71.4 ± 0.48 | 85.4 ± 0.33 |
| Duplicates, % | **24.30 ± 0.45** | 48.70 ± 0.12 | 27.4 ± 0.37 |
| Target coformers, % | **5.63 ± 0.22** | 2.14 ± 0.19 | 0.34 ± 0.05 |
| Diversity of target | 0.32 ± 0.02 | 0.24 ± 0.01 | **0.21 ± 0.01** |

While GPT-2 produced a higher percent of target coformers compared to T-VAE (2.14% versus 1.68%), the ensemble of generative models in GEMCODE resulted in at least two times more molecules with the desired physicochemical properties (5.63% with T-CVAE only). Llama-3-8B was able to generate only 0.34%, which is not attractive given vast amount of resources required to train and use the model for generation.

Meanwhile, Llama-3-8B clearly outperformed GPT-2 in terms of validity, novelty and the number of duplicates in molecule generations. It also showed almost on-par performance with GEMCODE, e.g., the percent of valid molecules (98.3%) was comparable to T-VAE (99.7%) and T-CVAE (98.4%). One important advantage of Llama-3-8B over GEMCODE is its ability to generate more diversity in the target molecular space. However, this advantage is diminished by the overall low percent of target coformers generated.

# E   Evolutionary optimization

## E.1   Framework

To find molecules with a higher probability of exhibiting desired characteristics, we used an evolutionary algorithm based on a self-developed GOLEM framework [85] [5]. The evolutionary algorithm operates on graph representation of molecules. The problem being solved is the minimization of the multi-objective $F$ in the discrete space of structural graphs $\mathbb{M}_{\mathfrak{d}}$ under a set of constraints $\mathbb{C}$. The task is to find an optimal structure $M_g^* = \langle V, E \rangle$.

$$M_g^* = \underset{\mathbb{M}_{\mathfrak{d}}}{argmin}\, F, \quad where \quad \mathbb{M}_{\mathfrak{d}} = \{M_g \mid \mathbb{C}(M_g)\}.$$

$$F = (1 - p_u(x), 1 - p_o(x), p_h(x))^T,$$

where $x$ is an evaluated molecule of coformer, $p_u(x)$ is the probability of the positive class for unobstructed planes, $p_o(x)$ is the same probability for orthogonal planes, and $p_h(x)$ – for H-bond bridging. Therefore, minimization of the fitness function $F$ leads to generation of coformer molecules having an improved tabletability profile.

## E.2   Scheme of the algorithm

The general scheme of the evolutionary algorithm is presented in the Figure 9. At first, individuals from an initial population are selected for mutation. At the mutation stage, the individuals are modified using a set of mutations described earlier. To control the process, Mutation operator refers to Change Advisor that determines possible actions. Since the algorithm implements a generational evolution scheme, Inheritance produces a new population using all the individuals obtained through mutation. Ellitism operator replaces the four worst individuals in the new population by the best ones found so far. The cycle is repeated until any stopping criteria are satisfied (time limit or maximal number of iterations).

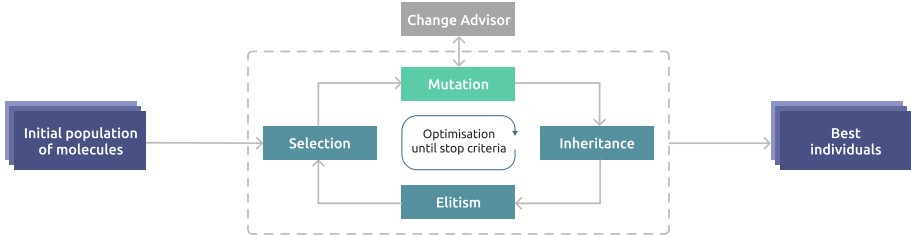

Figure 9: Scheme of the evolutionary algorithm that is used for fine-tuning of solutions.

## E.3   Evolutionary optimization experiment details

For each generative model, 10 independent runs of optimization were performed. For each run, we used a random sample from all unique coformers generated by the model as the initial population. The sample size was equal to the mean number of target coformers found in the 10,000 generated molecules. Analyzing the dataset of already known coformers [33], we estimated the maximal number of heavy atoms to be 50 and the available elements to be C, N, O, F, P, S, Cl, Br, and I. Those estimates were used to configure the evolution process. Population sizes were set to 200, number of iterations to 200 and timeout to 60 minutes.

Evolutionary algorithms tend to produce redundantly complicated structures due to overfitting [115]. To avoid unrealistic molecules, synthetic accessibility score (SA) [116] was calculated for all molecules obtained with evolutionary optimization. Only coformers with $SA \leq 3$ were selected for further consideration.

---

[5] https://github.com/aimclub/GOLEM

### E.4 Evolutionary optimization significantly improves H-bond bridging

The results of the evolutionary optimization application were the most prominent for H-bond bridging and are presented in the Figure 10.

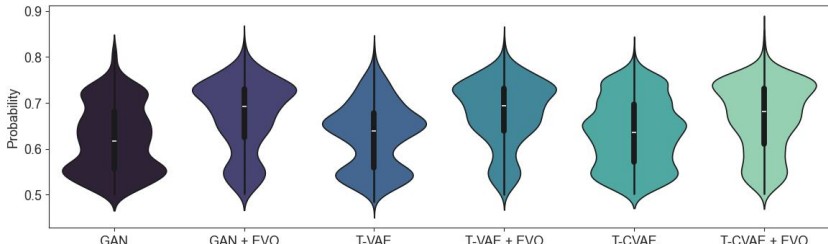

Figure 10: Comparison of probability distributions for the presence of hydrogen bonds between the planes (H-bond bridging) for coformers generated by the neural models and optimized by evolution.

In Figure 11 mean convergence of the algorithm is shown. Interestingly, the use of molecules generated by T-CVAE as an initial population also consistently increased the algorithm convergence speed.

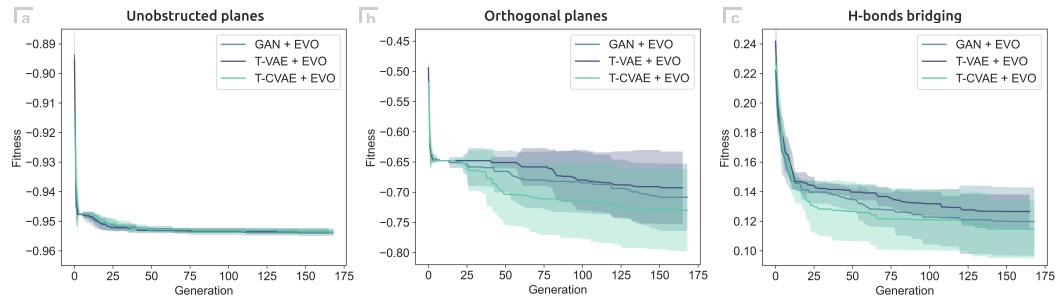

Figure 11: Convergence for mechanical properties of evolution starting from co-crystals generated by different models.

### E.5 Evolutionary schemes comparison

Table 7 presents results of comparison of two evolutionary schemes: SPAE-2 and MOEA/D.

Table 7: Results and statistical significance (non-parametric one-sided Mann-Whitney test) for 10 runs of evolutionary algorithms based on SPEA-2 and MOEA/D selections.

| Model | Property | Median probability | | p-value |
| | | SPEA-2 | MOEA/D | |
| --- | --- | --- | --- | --- |
| GAN | Unobstructed planes | 0.819 | 0.819 | 2,639 |
| | Orthogonal planes | 0.385 | 0.389 | 0.141 |
| | H-bond bridging | 0.692 | **0.694** | **0.039** |
| T-VAE | Unobstructed planes | 0.825 | 0.825 | 7,958 |
| | Orthogonal planes | 0.393 | 0.398 | 0.753 |
| | H-bond bridging | 0.693 | 0.692 | 8.703 |
| T-CVAE | Unobstructed planes | 0.826 | 0.823 | 1.049 |
| | Orthogonal planes | 0.387 | 0.390 | 0.137 |
| | H-bond bridging | 0.682 | **0.693** | **0.013** |

### E.6 Comparison with GraphGA baseline

The genetic algorithms are considered to be a strong basis for drug design tasks[53, 117], so we compare the GEMCODE with genetic baselines. Known baselines from *Guacamol*[118] can be used to optimise any molecule in SMILES notation with a given goal. However, unlike the Guacamol tasks, the co-crystal design task is multi-objective, whereas the algorithms from Guacamol_baselines[6] (e.g. the well-known GraphGA) are focused on single-objective tasks.

We have developed the multi-objective modification of GraphGA with Pareto dominance-based fitness, which can be used for the co-crystal design tasks. It started from a random subset of co-crystals (with the same population size and number of iterations as used in GEMOL).

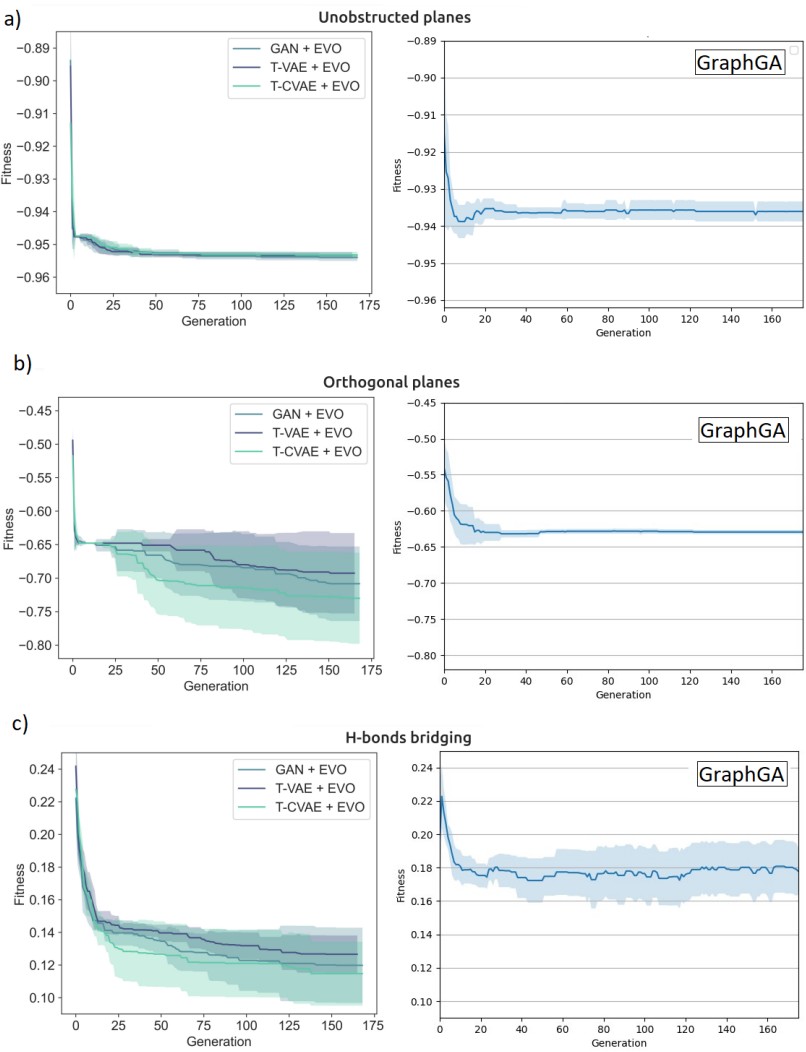

Figure 12: Comparison of GEMOL with GraphGA baseline. GEMCODE in the left, GraphGA in the right

GraphGA showed inferior results according to the mechanical property values obtained from GEMOL (see Figire 12). We can see that for unobstructed planes the best probability average over all runs is 0.95 for GEMCODE against 0.938 for GraphGA, for orthogonal planes - 0.72 against 0.63 and for h-bonds bridging - 0.18 against 0.12. Also, the convergence is quite unstable (we think it is caused by insufficiently successful selection procedure), so the hybrid approach implemented in GEMCODE is better in all cases.

---

[6] https://github.com/BenevolentAI/guacamol_baselines

Furthermore, the proposed hybrid approach is able to generate on average 21.3% of target molecules from the available population during optimisation. Our comparative tests for GraphGA showed 20.5% of targeting molecules from the population (with worse quality according to predicted mechanical properties). Finally, it should be noted that our existing dataset of 4223 coformers can be extended by a further 2452 molecules using a generative model (while GraphGA itself only provides 120 new candidate molecules).

# F Indicators

## F.1 Additional results on evolutionary optimization

Table 8 provides additional statistics on the impact of evolutionary optimization.

Table 8: Results and statistical significance of the evolutionary optimization. Mean and standard deviations are given for the probabilities of Unobstructed planes, Orthogonal planes and H-bond bridging.

| Model | Property | Probability | | p-value | Novelty |
|-------|----------|-----------|-----------|---------|---------|
| | | Generated | Optimized | | |
| GAN | Unobstructed planes | $0.82 \pm 0.05$ | $0.82 \pm 0.06$ | - | 0.68 |
| | Orthogonal planes | $0.39 \pm 0.04$ | $\mathbf{0.40 \pm 0.05}$ | 3.83e-12 | |
| | H-bond bridging | $0.62 \pm 0.07$ | $\mathbf{0.67 \pm 0.08}$ | 1.50e-67 | |
| T-VAE | Unobstructed planes | $0.82 \pm 0.05$ | $0.82 \pm 0.06$ | - | 0.72 |
| | Orthogonal planes | $0.39 \pm 0.05$ | $\mathbf{0.41 \pm 0.05}$ | 3.87e-10 | |
| | H-bond bridging | $0.63 \pm 0.07$ | $\mathbf{0.68 \pm 0.07}$ | 2.51e-66 | |
| T-CVAE | Unobstructed planes | $0.81 \pm 0.06$ | $\mathbf{0.82 \pm 0.06}$ | 1.36e-05 | 0.60 |
| | Orthogonal planes | $0.39 \pm 0.05$ | $\mathbf{0.40 \pm 0.05}$ | 2.69e-10 | |
| | H-bond bridging | $0.64 \pm 0.07$ | $\mathbf{0.67 \pm 0.08}$ | 2.60e-47 | |

We used multiple indicators to assess the performance of the generative models. Let us denote the number of generated molecules by $G$. All the generated molecules contain valid molecules $V$, duplicates $D$, new molecules that are not contained within the training dataset $N$, as well as molecules possessing the desired physicochemical properties $C$ and molecules satisfying the condition of synthetic accessibility molecules $S$ ($SA <= 3$). With these notations, we define quality indicators in the subsequent subsections: F.2-F.7.

## F.2 Validity

Validity refers to the ratio of predicted molecules deemed chemically plausible and estimated by `rdkit.Chem.MolFromSmiles` taking into account the valence of atoms in the molecule and the consistency of bonds in aromatic rings. When evaluating the validity of molecules using the `rdkit` package, we obtain a set of molecules equal to $V$. Then Validity can be calculated as the ratio of valid molecules to all generated molecules:

$$\text{Validity} = \frac{V}{G} \cdot 100 \ [\%].$$

## F.3 Duplicates

Duplicates is a ratio that shows how many duplicates are contained within all valid molecules:

$$\text{Duplicates} = \frac{D}{V} \cdot 100 \ [\%].$$

### F.4 Novelty

Novelty is a ratio that shows how many valid molecules without duplicates among the generated ones are novel (i.e., these molecules were not contained within the training dataset and were produced by the generative model). Therefore, Novelty is defined as follows:

$$\text{Novelty} = \frac{N}{V} \cdot 100 \ [\%].$$

### F.5 Target coformers

After we obtain a set of valid molecules without duplicates that are not contained within the training dataset, we predict the physicochemical properties using the pretrained gradient boosting model. We then select molecules by the required mechanical properties: Unobstructed planes $= 1$, Orthogonal planes $= 1$, H-bonds bridging $= 0$. For those, we also evaluate the synthetic accessibility ($SA$) score using `rdkit.Contrib.SA_score`. As mentioned before, we define the target tabletability profile in the generation process by the required mechanical properties and $SA \leq 3$. Thus, the percentage of target molecules is calculated as follows:

$$\text{Target coformers} = \frac{S}{G} \cdot 100 \ [\%].$$

### F.6 Synthetic Accessibility Score (SA)

$SA$ is calculated using `rdkit.Contrib.SA_Score`.

### F.7 Diversity

We used Diversity to assess the diversity of the generated molecules. In order to estimate this indicator, we need to calculate Tanimoto-similarity ($T_s$) and Tanimoto-distance ($T_d$). Thus, to estimate $T_s$, consider two molecules, $a$ and $b$, with Morgan fingerprints $m_a$ and $m_b$, respectively. The number of common fingerprints between the two molecules is represented by $|m_a \cap m_b|$, and the total number of fingerprints is represented by $|m_a \cup m_b|$. Then, Tanimoto-similarity is defined by:

$$\text{T}_s = \frac{|m_a \cap m_b|}{|m_a \cup m_b|}.$$

Then, Tanimoto-distance and Diversity are related as follows:

$$\text{Diversity} = T_d(a, b) = 1 - T_s.$$

## G   ML model

### G.1   Dataset splitting

When it comes to dataset splitting, the choice of technique has a significant impact on model performance. For the field of co-crystals, there is still no established splitting strategy. The potential for more stratified splitting approaches in drug design, such as molecular scaffolds, exists, but adapting this method to co-crystal dataset presents a number of challenges. Each sample in the co-crystal dataset consists of two coformer molecules with different scaffolds and a large structural diversity, unlike drug design applications that deal with specific classes of compounds and typically allow the identification of fewer scaffolds. For out-of-distribution generalisation analysis, along with random splitting of the dataset, we separated the training and test samples based on Tanimoto similarity, maximising dissimilarity between molecules of different subsets. The results of the experiment are summarised in Table 9. Since random splitting has already been used in the co-crystal domain within other works, it is practical and showed the best performance the random splitting was preferred.

Table 9: Metrics of the Gradient Boosting model for predicting the mechanical properties of co-crystals upon changing the data splitting strategy.

| Property | Random | | Tanimoto Similarity | |
|---|---|---|---|---|
| | Accuracy | F1 Score | Accuracy | F1 Score |
| Unobstructed planes | **0.73** | **0.77** | **0.73** | 0.71 |
| Orthogonal planes | 0.79 | **0.59** | **0.83** | 0.52 |
| H-bonds bridging | **0.73** | **0.76** | 0.70 | 0.70 |

## G.2 Threshold

Due to the significant imbalance of data related to the orthogonal planes, we decided to change the threshold of probability of assigning a sample to a class. For this purpose we investigated the change of precision and recall depending on the value of the introduced threshold (Figure 13).

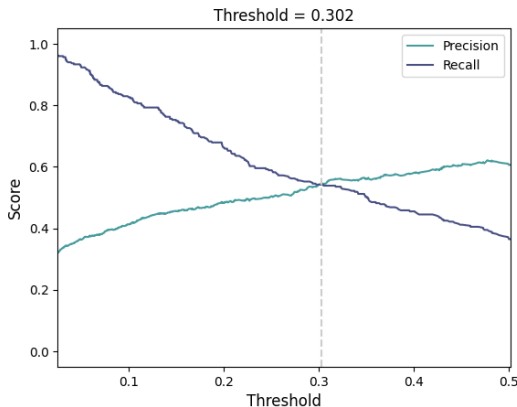

Figure 13: Change of metrics depending on the set threshhold.

The threshold was set under the condition of equality (intersection of lines) of precision and recall metrics, as it represents the optimal point for balancing the number of false positives and false negatives.

## G.3 SHAP analysis

In order to increase the transparency and reliability of the ML model's decisions and results, we used the SHAP method, which allows us to interpret the output of the predictive model. But also, it enables domain-specific hypothesis generation while contributing to the explainability of the predictive model, which is a huge benefit for potential applications. The Figure 14 shows the features that were used in the training process, ranked in order of importance for the final prediction. In this case, the SHAP values to the left of the center vertical line are negative-class (0) and to the right are positive-class (1). Also, red dots indicate a higher feature value and blue dots indicate a lower feature value.

## G.4 Prediction of co-crystals properties

One of the interesting applications of machine learning models in the field of co-crystal design is the prediction of different physicochemical properties. Table 10 presents a comparison of known models that predict properties such as crystal density, entropy and enthalpy of melting, melting temperature, ideal solubility, and lattice energy. It is important to note that the mechanistic properties of co-crystals have not been predicted before, so the metrics obtained in this paper can be considered state-of-the-art.

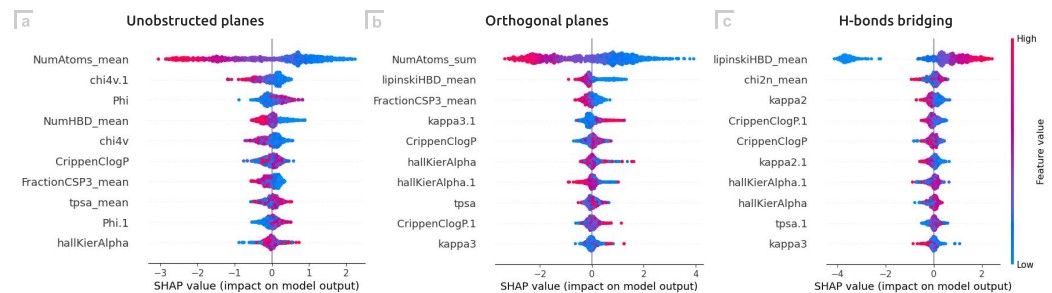

Figure 14: SHAP plots demonstrating the importance of the first 10 coformer features for prediction of (a) Unobstructed planes, (b) Orthogonal planes and (c) H-bond bridges.

Table 10: Comparative table with model metrics on prediction of various co-crystals properties.

| Task | Property | Data points | Best metric | Generative design | Ref |
|---|---|---|---|---|---|
| Regression | Crystal density | 26 | $R^2 = 0.993$ | No | [64] |
| Regression | Melting temperature | | $R^2 = 0.992$ | No | |
| Regression | Melting enthalpy | | $R^2 = 0.999$ | No | |
| Regression | Melting entropy | 30 | $R^2 = 0.997$ | No | [35] |
| Regression | Ideal solubility | | $R^2 = 0.953$ | No | |
| Regression | Melting temperature | 61 | RSD = 2.89% | No | [62] |
| Regression | Lattice energy | | RSD = 2.40% | No | |
| Regression | Crystal density | 61 | RSD = 1.77% | No | [63] |
| Regression | Melting temperature | 84 | $R^2 = 0.998$ | No | [65] |
| Regression | Crystal density | 4144 | $R^2 = 0.985$ | No | [36] |
| Classification | Unobstructed planes | | Accuracy = 0.731 | | |
| Classification | Orthogonal planes | 6029 | Accuracy = 0.785 | Yes | Our work |
| Classification | H-bonds bridging | | Accuracy = 0.734 | | |

## G.5 AutoML

To prove the effectiveness of the proposed ML models, we conducted additional experiments with the state-of-the-art AutoGluon [7] framework. We used the timeout of one hour and the "best quality" preset. After extensive evaluation, we were able to achieve no significant improvement of the F1-score against the proposed model (Table 11). These results indicates that the ML models in GEMCODE are less prone to overfitting and have better generalization capabilities.

Table 11: Comparison of the proposed ML models with AutoML. Best achieved metrics are given.

| Property | Model | Precision | Recall | F1 Score |
|---|---|---|---|---|
| Unobstructed planes | AutoGluon | **0.74** | **0.73** | **0.72** |
| | Our model | 0.73 | **0.73** | **0.72** |
| Orthogonal planes | AutoGluon | 0.78 | **0.79** | 0.78 |
| | Our model | **0.79** | **0.79** | **0.79** |
| H-bond bridging | AutoGluon | **0.81** | 0.71 | 0.68 |
| | Our model | 0.77 | **0.72** | **0.71** |

---

[7] https://github.com/autogluon/autogluon, version 1.1.0

