# OpenReview forum: "Hybrid Generative AI for De Novo Design of Co-Crystals with Enhanced Tabletability"
_NeurIPS.cc/2024/Conference — NeurIPS 2024 poster_

### Official Review · Reviewer_qBrF · 2024-07-03

**Soundness:** 2
**Presentation:** 3
**Contribution:** 2
**Rating:** 5
**Confidence:** 3

**Summary:**

This paper presents GEMCODE, the first co-crystal design AI pipeline, which consists of four components:
- SMILES-based models for coformer generation.
- Classification models for co-crystal property prediction.
- An evolutionary algorithm for coformer optimization.
- A GNN for prediction of the probability of co-crystal formation.

In addition, experiments are carried out on each component and the entire pipeline.

**Strengths:**

This paper is the first to introduce generative AI into co-crystal design, which is a very important topic for drug development and other fields. Specifically, this paper establishes a complete framework for co-crystal design, including datasets, property prediction, coformer generation, optimization, and validation, which can be recognized as the initial baseline in this field. In addition, the experimental results are detailed and the logic is clear.

**Weaknesses:**

1. The experimental results of property prediction show that the three mechanical properties are inherently elusive, because the accuracy and F1 score of most models in Figure 2 are less than 0.8, which is not ideal for binary classification tasks. Therefore, my concern is that this paper is a pioneer in both the property prediction and generation/optimization of co-crystals, and will the defects (or biases) of the property prediction models be brought into the generation/optimization?

2. In GEMCODE, coformer de novo generation and optimization are divided into two steps, which I think is to follow the tradition in the field of drug discovery. However, in drug discovery, the main purpose of de novo generation is to find molecular candidates whose "main properties" (such as docking scores) are satisfactory, while the main purpose of optimization is to optimize some other properties (such as toxicity and solubility) while maintaining the "main properties" (usually by a similarity constraint). The property objectives of de novo generation and optimization in GEMCODE is the same (without any similarity constraint), which makes me doubt the rationality of the pipeline.

3. Admittedly, I am not familiar with the co-crystal's chemical background, so it is difficult for me to evaluate GEMCODE's chemical validity. Actually, I also did not find any paper on co-crystals in top conferences on machine learning, which makes me question this paper's suitability for publication at this conference. Overall, the main contribution of this paper is to model the computational pipeline of co-crystal design, and to apply existing AI techniques to it. Therefore, in terms of style, I think the chemistry and cheminformatics community may be more appropriate for this paper than a machine learning conference.

**Questions:**

1. In Table 1, the "diversity of targets" are all greater than 0.9. I'm not sure if they are values for internal diversity, and if they are, the diversity values are quite large for sets containing hundreds of molecules. From a molecular design perspective, this suggests that the design objective may be so simple that many very dissimilar molecules fit it.

2. In the paper, the tabletability of co-crystals is presented as a target to be improved. However, in GEMCODE, the three properties related to tabletability are all binary variables, so for a coformer molecule, its tabletability is also represented by one binary variable. Therefore, is the word "enhance" inappropriate for tabletability?

3. Of the three coformer molecules generated in Table 3, two contain ions. I'm not sure if ions are common in coformer molecules, but I think they're not common in the ChEMBL database. So, is there a significant distribution difference between the pre-trained and fine-tuned datasets?

**Limitations:**

Nothing beyond what has already been stated in Section 6.

---

> ### Author Rebuttal · Authors · 2024-08-07
>
> We appreciate the reviewer's useful comments and suggestions!
> Below, we would like to provide our __answers to the questions__:
>
> 1. We are happy to provide a clarification for the high values of "Diversity of target" in Table 1. The diversity values were calculated for the molecules that have been pre-screened for novelty (meaning they are not present in the training dataset), validity, and matching mechanical properties. Therefore, a diversity value exceeding 0.9 appears to be reasonable. Furthermore, the formation of co-crystals can involve molecules with varying chemical structures. Consequently, GEMCODE should have the capability to accommodate diverse chemical structures. This is achieved through hybridization of generative neural networks with evolutionary optimization.
> 2. Regarding tabletability, we believe that creating a coformer molecule with a target tabletability profile (i.e., the target mechanical properties) can be seen as an enhancement in the tabletability of the co-crystal. However, we are open to adjust the wording if the reviewer suggests a more accurate term.
> 3. To observe the difference between the pretraining and the fine-tuning datasets, please refer to the t-SNE visualization in Figure 5b of Appendix D.2. The figure shows that the coformer dataset produces a more concentrated set of molecules compared to ChEMBL. The list of coformers was sourced from [1], which was also used by the CCGNet model for predicting co-crystallization probabilities. By integrating CCGNet and training all the other components of our pipeline on the same data, we show consistency of the GEMCODE design. The presence of ionized forms in Table 3 arises from SMILES representations in the coformer dataset indicating charge distribution within _neutral_ molecules (e.g., "Nc1nc2ccc(N+[O-])cc2s1" with a nitro group,  "CSCCC([NH3+])C(=O)[O-]" with carboxyl and amino groups). Consequently, the model has learned to generate charged molecules (e.g., with carboxyl "C(=O)[O-]") in certain instances. We are expanding the scope of GEMCODE beyond co-crystals to encompass other crystalline forms, such as salts. This expansion will aim to specifically address model biases related to assigning charged molecules to co-crystals.
>
> We would like to also __comment on the weaknesses__ outlined by the reviewer:
>
> 1. We thank the reviewer for highlighting a key limitation in our study. To our knowledge, we are the first to predict mechanical properties of organic co-crystals. Therefore, our work sets the state of the art for the problem. We have run extensive evaluations of various machine learning models incorporating a range of descriptors striving for better performance. We are confident that the capability of predictive models is limited by the training data. As more data of sufficient quality becomes available, we will refine GEMCODE accordingly. Notably, despite the current limitations, several validation cases presented in this work demonstrate the capability of GEMCODE to successfully predict new co-crystals, experimentally validated and reported in the literature.
> 2. The main motivation for incorporating evolutionary optimization in GEMCODE was to address the aforementioned data limitations. Our dataset consists of approximately 6000 coformers. Training generative neural networks on a dataset of this size may lead to a restricted diversity in the generated molecules. By implementing evolutionary optimization as a separate step in the pipeline, we overcome this limitation by design, as the evolutionary algorithms operate independently of the training data. Furthermore, evolutionary optimization can help mitigate the drawbacks of the machine learning models discussed earlier. The optimization process focuses on enhancing the likelihood of coformers possessing all the necessary mechanical properties. As a result, a molecule generated by the neural networks can be further refined through evolutionary optimization in terms of this likelihood. In such a scenario, the evolutionary optimization step is complementary to the initial generation and demonstrates the advantage of the hybridization approach we presented. Finally, evolutionary optimization is designed to retain the coformers with the highest likelihood on each iteration. In other words, it is guaranteed to improve the mechanical properties of coformers while adding diversity to the pool of the generated molecules.
> 3. We are confident that our paper aligns well with the NeurIPS guidelines being an _application_ of _machine learning for sciences_ (https://neurips.cc/Conferences/2024/CallForPapers). While co-crystals may not have been a prevalent topic at A* conferences, we see this as an opportunity to expand the horizons of machine learning research and engage with researchers from other fields. Notably, a recent study in this direction was presented for the first time at the ICML 2024 workshop [2]. This only highlights the originality and significance of our work.
>
> In conclusion, we would like to quote the reviewer, _“this paper is a pioneer in both the property prediction and generation/optimization of co-crystals”_. Although this was expressed as a concern, we would like to point out that this quote, in fact, is highlighting a great scientific achievement. Provided that we have sufficiently addressed all the comments and questions, we kindly ask the reviewer to consider increasing the rating to 6. Given that we have initially received the ratings of 8/6/4/4, this would significantly increase our chances to get accepted.
>
> __References:__
>
> [1] Jiang, Y., Yang, Z., Guo, J., Li, H., Liu, Y., Guo, Y., ... & Pu, X. (2021). Coupling complementary strategy to flexible graph neural network for quick discovery of coformer in diverse co-crystal materials. Nature Communications, 12(1), 5950.
> [2] Birolo, R., Özçelik, R., Aramini, A., Gobetto, R., Chierotti, M. R., & Grisoni, F. (2024). Deep Supramolecular Language Processing for Co-crystal Prediction.

---

> > ### Comment · Reviewer_qBrF · 2024-08-10
> >
> > Thank you for your detailed rebuttal and for addressing the concerns raised. I decide to raise my score up to 5.

---

### Official Review · Reviewer_3KPG · 2024-07-08

**Soundness:** 3
**Presentation:** 4
**Contribution:** 4
**Rating:** 8
**Confidence:** 4

**Summary:**

This works presents a generative framework for co-crystal design which uses deep learning and evolutionary algorithms for optimization. The GEMCODE pipeline can be used to select optimal molecular pair combinations: an active pharmaceutical, and a coformer to control for the desired co-crystal properties. The work is an application of generative AI (specifically, GANs and VAEs) to an interesting and important problem in the pharmaceutical industry. The paper has great figures and is well-written, and the code appears well-documented. Nevertheless, the paper could improve from some slight re-structuring and/or re-writing to improve the clarity of the paper, as it was tough to follow in places (there are many molecular representations and model architectures used, many target properties, these could be better organized so that it is easier to follow). But overall I really enjoyed reading it and would like to congratulate the authors on the nice work.

**Strengths:**

* The work makes extensive use of existing frameworks (e.g., CCGNet) and datasets (e.g., ChEMBL, CSD) where possible, which is great, while still exploring the development of new tools when it comes to the generative aspect.
* Overall the study is very thorough - many of the questions I found myself having while reading the paper were either answered later on in the paper or in the appendix. Those which were not, I have written below.
* Good details provided in the appendix for the data and many of the methods, it seems it should be reproducible from the details provided herein (as well as the linked code).
* This paper highlights an important application of generative AI to a new domain.

**Weaknesses:**

* It was a good choice, nonetheless, to explore non-neural models for the prediction of mechanical properties, which is where the authors have the fewest data points (6K). However, given the limited data on coformers (about 7K co-crystals), I also would have expected a simpler baseline like a random forest for estimating likelihood of co-crystal formation.
* Clarity could be improved in places, since the authors are working with lots of different representations and models throughout their pipeline. Perhaps this could be better visualized in a way that gives a quick overview of the data type, number of data points, and model architecture, for each model included in the pipeline.
* The model may have benefited from a more advanced hyperparameter tuning for the generation of coformers (e.g., Optuna rather than grid search), as the models and their final performance can be quite sensitive to the selected hyperparameters. For instance, %validity >99% should be attainable in all cases for the molecular generative models presented here, to which the models were close but not all quite there.
* Some of the notation is not well-explained. For instance, in the target coformers equation, what is S? In many of the tables and figures, the axes labels or certain notation is implied, which may be mentioned somewhere else in the paper, but it would improve readability if readers could have a reminder of what that property is (and if it is bounded, a percentage, etc). Also, where there are error bars, it should be stated what these are (e.g., in tables, figures). It is a long paper so this would be appreciated, I found myself flipping back and forth a lot.
* To better understand the diversity of the chemical space spanned by generated molecules, it would have been interesting if the authors better quantified that of the training set as well, and if different training/testing strategies could better assess the generalizability to new coformers rather than simply splitting the data. Were not certain classes of coformers more represented than others?
* Given that the authors want others to use their pipeline to discover new coformers, it would have been great to quantify the coverage of chemical space of the model by metrics other than % validity and % novelty. Did the authors consider such metrics (or even dimensionality reduction visualization/techniques) to get a better sense of the chemical space coverage?

**Questions:**

* When predicting co-crystal formation, do the authors also have any negative pairs, e.g., molecules that definitely do not co-crystalize? I would assume the bias is towards positive pairs, which perhaps leads to overestimation of the likelihood of co-crystallization for any new pair of molecules, but this was not fully clear to me.
* In the validation case studies (section 5.4), it is unclear if molecules with overlap or significant molecular similarity to nicorandil, rivaroxaban, and paracetamol were excluded from the training set before using the model for these tasks, otherwise there may be data leakage and the conclusions are not so meaningful.
* I see that molecular descriptors (I am guessing 2D?) were used for representing the coformers for mechanical property prediction. Were the fingerprints also explored here, to less effectiveness? Curious since it seems fingerprints were used in other parts of the pipeline, and not sure if descriptors were necessary for good performance here or not.
* Related to the above question, how important is 3D information? I would expect a lot, and wondering if the authors considered including 3D atomic-level descriptors as well in any of their models.

**Limitations:**

* In addition to the final median probability scores attained by each model for the molecular optimization tasks explored in this study, it would be relevant and interesting to report the sample efficiency. This is a much more relevant metric in this context of molecular optimization, rather than the final probability score, as it gives a measure of how quickly a model learns (e.g., how many oracle calls are needed to obtain a specific score). Recommend to include an estimate of the sample efficiency in a revised version.
* Is the “median probability score” not in fact a likelihood of sampling specific coformers (and not really a probability)? If so (or if not), I think this could be presented more clearly, as what exactly this metric is was not clear to me. It was probably defined somewhere in the paper but I could not find it easily.
* One big limitation is that it would have been relevant to compare the models for the coformer optimization tasks to a simple baseline, such as a “virtual screening” of the coformers. The advantage of using the generative model presented herein should be that better molecules, and thus better scores, should be achievable with the generative model than by simply screening the database (given a fixed sample budget), but this has not been demonstrated in the current study. It would be a pretty simple/standard baseline since the reference database is small (~7K coformers).

---

> ### Author Rebuttal · Authors · 2024-08-07
>
> We thank the reviewer for the high rating and very valuable comments! We appreciate the detailed feedback on where we can improve the clarity of the manuscript. We will do so in the camera-ready submission.
>
> Below, we provide our __answers to the questions__:
>
> 1. For the purpose of ranking molecular pairs according to the probability of co-crystallization, we employed the Co-Crystal Graph Network (CCGNet) model [1]. The authors used a dataset of 6819 positive and 1052 negative samples to train the model, so there is indeed an imbalance problem. Nevertheless, CCGNet was shown to achieve high accuracy on negative samples (97.26%) thanks to the combined use of graph representations of the underlying GNN and 12 molecular descriptors.
> 2. In the validation case studies (Section 5.4), we predicted three co-crystal systems that were not present in the training dataset. By demonstrating that GEMCODE can generate novel co-crystal structures reported in the literature as experimentally validated, we aimed to showcase its strong predictive capabilities. We understand the reviewer's concern about data leakage and would like to clarify that we ensured that none of these co-crystal systems were included in the training set prior to the validation process.
> 3. In our study, we indeed utilized a variety of descriptors to predict mechanical properties, encompassing molecular fingerprints of various types (Morgan, MACCS) and lengths (166, 512, 1024, 2048), molecular descriptors sourced from different origins (RDKit, Mordred, PaDEL), and 3D descriptors generated from RDKit (Autocorr3D, MORSE, PMI, etc.). We found that a set of molecular descriptors with physicochemical properties (29, 24, and 30 features for unobstructed planes, orthogonal planes, and hydrogen bonding, respectively) resulted in the best predictive performance.
>
> We would like to also __comment on the weaknesses__ outlined by the reviewer:
>
> 1. We appreciate the suggestion to consider using more advanced methods like Optuna for hyperparameter optimization. While we acknowledge that Optuna may provide more efficient tuning compared to grid search, we would like to highlight that grid search is a pragmatic choice for a moderate number of experiments (10-20), delivering acceptable results. In this work, we had to investigate and optimize numerous configurations of the pipeline. Given the complexity and time constraints associated with more advanced methods, we believe that our current approach is a tradeoff between performance and computational cost. We leave the more advanced methods for GEMCODE hyperparameter optimization for the future work.
> 2. To our knowledge, there is no common stratification strategy for splitting the data in this domain. Another study on co-crystals accepted to ICML 2024 also employed a standard random split methodology [2]. Based on our empirical results, we do not expect any significant change in results for alternative data splits.
>
> In addition, we would like to __comment on the limitations__ pointed out by the reviewer:
>
> 1. Regarding the "median probability score", we believe it can be best described as the median probability of assigning coformers to a positive class for each of the mechanical properties. In other words, median probability score gives an idea about the central tendency of the model's confidence in predicting a particular mechanical property. This is, of course, related to the likelihood but reflects a different aspect of the model behavior. We will make sure to clarify this further in the camera-ready version of the manuscript.
> 2. Table 1 illustrates the comparison of generative models focusing on “Target сoformers”. The percentage indicated in this row represents the ability of the model to generate new coformers with the desired co-crystal properties that were not present in the training dataset. We investigated that, for the Theophylline drug, the percent of target coformers (satisfying all three mechanical properties) in the training set was only 8.46%. Generating candidate coformers for this drug with T-CVAE only, we obtained 6.52% of target coformers. This clearly demonstrates how GEMCODE is capable of discovering previously unknown co-crystal systems. We appreciate the suggestion to illustrate how probability distributions of mechanical properties compare between the training and the generated data. We will add this evaluation to the camera-ready submission.
>
> __References:__
>
> [1] Jiang, Y., Yang, Z., Guo, J., Li, H., Liu, Y., Guo, Y., ... & Pu, X. (2021). Coupling complementary strategy to flexible graph neural network for quick discovery of coformer in diverse co-crystal materials. Nature Communications, 12(1), 5950.
> [2] Birolo, R., Özçelik, R., Aramini, A., Gobetto, R., Chierotti, M. R., & Grisoni, F. (2024). Deep Supramolecular Language Processing for Co-crystal Prediction (https://openreview.net/forum?id=bQ9d2hzjW4&noteId=9SoErgR0kb).

---

> > ### Comment · Reviewer_3KPG · 2024-08-08
> > **Reviewer Response**
> >
> > Thank you to the authors for the detailed response and for the clarifications.
> >
> > One thing that I disagree with still is the comment that
> > > To our knowledge, there is no common stratification strategy for splitting the data in this domain... Based on our empirical results, we do not expect any significant change in results for alternative data splits.
> > I do not think this is something that should be swept under the rug. Doing a random split is a standard way of assessing in-distribution generalization, but can you think of other ways to split the data to asses out-of-distribution generalization? I think there are a few interesting experiments you could do, which are not difficult, that would add immensely to the value and utillity of the work. These are also standard experiments to run in applied ML papers.
> >
> > Related to the above point, there also seems to in general be a lot of confusion about what properties/features are spanned by the data (looking at the other reviewer comments as well as mine), which I think could be better clarified.

---

> > > ### Author Response · Authors · 2024-08-08
> > >
> > > We appreciate the reviewer's prompt reply and the chance to provide additional clarifications. We also fully agree that one should not neglect the importance of data splitting and sufficient empirical evidence supporting such study design choices should be provided. We are aware of molecular scaffolding as a method commonly used in drug design to evaluate out-of-distribution generalization capability of predictive models. However, adapting this method to our co-crystal dataset presents several challenges. First, each sample consists of two coformer molecules with different scaffolds (https://anonymous.4open.science/r/GEMCODE/rebuttal/cocrystals.png). Second, the coformers can exhibit a variety of structures, resulting in a large number of selected scaffolds for analysis (approximately 1000 scaffolds based on our preliminary experiments using the Murcko decomposition method). In contrast, in drug design applications (https://greglandrum.github.io/rdkit-blog/posts/2024-05-31-scaffold-splits-and-murcko-scaffolds1.html), researchers typically work with specific compound classes where it is feasible to identify a relatively small number of scaffolds (10-20). Therefore, given the moderate size of the co-crystal dataset, scaffolding might not produce statistically significant evaluations.
> > >
> > > At the moment, we identify another  approach as the most promising to further shed light on out-of-distribution generalization. In essence, the data can be split based on Tanimoto Similarity, maximizing dissimilarity between the molecules of different subsets. We are going to conduct such experiments in-depth and include the key findings to the Appendix of the camera-ready submission. We are thankful to the reviewer for rigorous evaluation of our work and attention to detail.

---

> > > > ### Comment · Reviewer_3KPG · 2024-08-13
> > > > **Reviewer Response**
> > > >
> > > > Thank you for the clarification. I can understand how the moderate size of the co-crystal dataset would make a scaffold split challenging.
> > > >
> > > > Splitting by Tanimoto similarity is a good idea, and will be a good experiment to do in addition to a random split, to better assess OOD generalization.

---

### Official Review · Reviewer_rNY2 · 2024-07-12

**Soundness:** 3
**Presentation:** 4
**Contribution:** 3
**Rating:** 5
**Confidence:** 3

**Summary:**

This paper presents GEMCODE, a novel pipeline for generating co-crystal designs with enhanced tabletability properties for pharmaceutical applications. The authors combine deep generative models, evolutionary optimization, and machine learning to create and evaluate potential co-former molecules. They train models to predict mechanical properties of co-crystals and generate co-former candidates using various approaches, including GAN, transformer-based VAE and CVAE architectures. The generated candidates are then optimized using evolutionary algorithms and ranked based on co-crystallization probability. The authors report that their T-CVAE model produced the highest percentage of co-formers with desired tabletability profiles. They validate their approach by generating experimentally confirmed co-formers for drugs like Nicorandil, Rivaroxaban, and Paracetamol. While the results appear promising, the authors acknowledge limitations such as potential bias in property predictions and the need for more comprehensive experimental validation. They also explore the use of language models for co-former generation, finding potential but noting the need for further optimization to achieve competitive performance.

**Strengths:**

I appreciate the comprehensiveness of the experimental evaluation in this paper. The authors have conducted a thorough examination of GEMCODE's performance across multiple dimensions. They systematically compared three different generative models (GAN, T-VAE, T-CVAE) using various metrics such as validity, novelty, and percentage of target co-formers generated. While the described methods are not new (models for property prediction, co-former generation, and co-crystal prediction), this is a novel task that the authors have decided to tackle with GEMCODE.

The authors evaluated the effectiveness of machine learning models for predicting mechanical properties of co-crystals, comparing performance before and after feature engineering. They assessed the impact of evolutionary optimization on the generated co-formers, providing statistical analysis of improvements in desired properties. The pipeline was validated using real-world case studies with known drugs, demonstrating its ability to generate experimentally confirmed co-formers.

**Weaknesses:**

I recommend adding these references in the introduction and related works, as they are relevant to the manuscript [1, 2, 3]. In table 1 and 2, arrows indicating the direction of optimization for the metrics would be helpful.

While the authors provide some validation using known co-formers, there is a lack of comprehensive experimental testing of the novel co-formers generated by GEMCODE. Synthesis and physical testing of predicted co-crystals would significantly strengthen the claims about the pipeline's effectiveness; or at least some theoretical ab initio results on the validation  More details on the validation experiments are needed, in particular how the therapeutic molecules were selected.

[1] Dollar, Orion, et al. "Attention-based generative models for de novo molecular design." Chemical Science 12.24 (2021): 8362-8372.
[2] Jensen, Jan H. "A graph-based genetic algorithm and generative model/Monte Carlo tree search for the exploration of chemical space." Chemical science 10.12 (2019): 3567-3572
[3] Tripp, Austin, and José Miguel Hernández-Lobato. "Genetic algorithms are strong baselines for molecule generation." arXiv preprint arXiv:2310.09267 (2023).

**Questions:**

Can the authors confirm if the validation experimentation generated co-formers are not present in the training set (the pre-training CHEMBL set or the fine-tuning co-former set)? Are the therapeutics chosen in the validation experiments also not present in any of the datasets (the co-former set in particular)?

The validation experiment needs to be expanded, as the results here would determine the utility of the entire GEMCODE pipeline. While the individual parts of the pipieline work, the validation experiments show that the co-crystal designs are real. Are there any fitness functions or ab initio results that can demonstrate the effectiveness of GEMCODE in generating co-crystals with novel therapeutics?

The state-of-the-art molecular generation algorithms [1], particularly for drug design tasks, are evolutionary. Rather than using language models, it seems more likely that a genetic algorithm would generate better structures than the GAN/VAEs/LLMs used here.

[1] Tripp, Austin, and José Miguel Hernández-Lobato. "Genetic algorithms are strong baselines for molecule generation." arXiv preprint arXiv:2310.09267 (2023).

**Limitations:**

The authors acknowledge a bias in predicting the absence of orthogonal planes, which could limit the pipeline's effectiveness. More work is needed to address this imbalance in the training data or model architecture. Furthermore, while tabletability is important, the narrow focus on this property overlooks other crucial aspects of pharmaceutical co-crystals such as solubility and bioavailability. This limits the utility of the pipeline for drug development. The authors acknowledge these issues in the limitations and conclusions.

---

> ### Author Rebuttal · Authors · 2024-08-07
>
> We thank the reviewer for the thoughtful evaluation of our work and for the valuable feedback! We will certainly revise and address all the comments and suggestions in the camera-ready version of the paper and our future work.
>
> Below, we provide __answers to the key questions and comments__:
>
> 1. We appreciate the suggestion to include references to additional articles on molecular generative design. We acknowledge the importance of providing a comprehensive overview of existing works in the field and will incorporate the suggested references in the Introduction and Related works sections.
> 2. We thank the reviewer for pointing out the need for arrows in Table 1 and Table 2 to simplify reading the metrics. We will make sure to incorporate this suggestion in the camera-ready submission.
> 3. We are currently focused on further validation of GEMCODE through experimental studies. To date, we have extensively validated each component of the pipeline as well as the predicted coformers of the case studies. The predicted coformers for Nicorandil, Rivaroxaban and Paracetamol (Section 5.4) were not present in the training data but these co-crystals are already experimentally confirmed, as reported in the literature. These cases provide strong evidence for the effectiveness of our pipeline.
> 4. For testing the pipeline, we selected those drugs reported to fail in forming tablets through direct pressing. Finding coformers for such drugs with GEMCODE was, in our opinion, the strongest evidence for the utility of the pipeline. We emphasize that our training data did not include any co-crystals related to Nicorandil and Rivaroxaban, but only systems of Paracetamol due to its widespread use. Therefore, Nicorandil and Rivaroxaban were entirely new to GEMCODE. This demonstrates the pipeline's ability to predict new coformers for both, new drugs and those contained in the training data.
> 5. We agree that many SOTA drug design approaches use evolutionary algorithms as optimizers (e.g. [1]). For this reason, we integrate them as part of our pipeline. However, the results of evolutionary optimization for co-crystals are highly dependent on the quality of the initial solution pool [2]. Furthermore, it may be too computationally expensive to wait for convergence starting from a random population. At the same time, restricting initial population to the existing database negatively affects the diversity of predicted solutions. Similar issues arise for other widely used molecular optimisation approaches. Therefore, we opted for a hybrid approach combining the strengths of design approaches of different nature.
> 6. To cope with the problem of unbalanced data, we experimented with multiple approaches on the data level (oversampling, undersampling and others) and on the model level (application of other models, adjustment of weights and others). The best metrics were achieved with the approach described in the paper using the threshold for the probability of the positive class. However, we plan to continue working towards improving the performance of the model in predicting orthogonal planes.
> 7. We appreciate the reviewer's valuable feedback regarding the importance of incorporating additional properties into our pipeline. In fact, we are actively working on extending our predictions to include solubility, tendency to form crystalline hydrates, and other properties of co-crystals. We are developing GEMCODE as an open source project, so we will release these updates as soon as they are sufficiently tested and validated. We believe that these enhancements will further improve the utility and applicability of our pipeline in the pharmaceutical and other relevant domains.
>
> We would like to ask the reviewer to consider raising the score to 7, due to the novelty and importance of the application problem that GEMCODE solves. As far as we know, no paper on this topic has been presented at A* conferences yet (except for the workshop at ICML 2024 [3]). We have done substantial work on GEMCODE development and now have a chance to pioneer the application of generative AI to organic co-crystal design. We will continue to work on GEMCODE (including experimental validation) and will definitely take into account the comments of all reviewers.
>
> __References:__
>
> [1] Ye Z. H. et al. Searching new cocrystal structures of CL-20 and HMX via evolutionary algorithm and machine learning potential //Journal of Materials Informatics. – 2024. – Т. 4. – №. 2.
> [2] O'Connor, D. (2023). (Co-) Crystal Structure Prediction With Machine Learned Potentials (Doctoral dissertation, Carnegie Mellon University).
> [3] Birolo, R., Özçelik, R., Aramini, A., Gobetto, R., Chierotti, M. R., & Grisoni, F. (2024). Deep Supramolecular Language Processing for Co-crystal Prediction (https://openreview.net/forum?id=bQ9d2hzjW4&noteId=9SoErgR0kb).

---

> ### Comment · Reviewer_rNY2 · 2024-08-09
>
> I thank the authors for their comprehensive rebuttal. The work done here is comprehensive and well motivated. And the author response has also sufficiently addressed my concerns.
>
> I believe the inclusion of some sort of theoretical validation of any future novel co-crystals would be highly valuable, if at all possible. Experimental validation is expensive, and the authors have demonstrated that previously reported literature results not in the validation set confirm that GEMCODE is effective. When open-sourced, having some sort of way to quickly provide theoretical estimates would be very useful for potential users; almost providing a virtual screening sort of pipeline. Perhaps even a regression model fitted to the available data to act as a proxy.
>
> I am willing to **increase review score to 7**.

---

> > ### Author Response · Authors · 2024-08-09
> >
> > We are very grateful to the reviewer for consideration and the decision to increase the score!
> > In the author console, we still see the rating of 6 though. It could be due to a technical issue with OpenReview. Please make sure to have edited and submitted the updated rating in the original review ("Edit" > "Official Review" > ... > "Submit"), such that it is reflected in the average rating of our work. Thank you!

---

### Official Review · Reviewer_XDyo · 2024-07-12

**Soundness:** 2
**Presentation:** 3
**Contribution:** 2
**Rating:** 4
**Confidence:** 5

**Summary:**

The authors investigate an interesting chemical problem of generating coformers given an organic molecule such that they would form co-crystals with desirable chemical properties. The authors use GAN/VAE-based methods to generate SMILES of potential coformers, which are then improved by evolutionary optimization, before finally predicting the probability of co-crystallization via a GNN. Traditional statistical models appear to be able to predict certain mechanical properties of the co-crystals given the pairs of SMILES. The GAN/VAE models are able to generate valid molecules, where ~40-80% are predicted to crystallize.

**Strengths:**

*Originality & Significance* The manuscript tackles a very interesting and impactful problem in pharmaceutical production, and overall a long-standing challenge in crystal engineering. To my knowledge, this is one of the first efforts to use generative ML to design co-formers. If completely new crystallization agents/co-formers can indeed be reliably produced, it would be a very useful tool.

*Quality*: In general, the approach taken here makes sense.

*Clarity*: The paper is very clearly written, and the presentation is easy to follow.

**Weaknesses:**

1. As with any ML for science work, the devil is in the details of the application. There are several big problems I see:
- Given that co-crystals are known to also have a variety of polymorphs, predicting mechanical properties based on SMILES without predicting/knowing the exact crystal structure inherently is not a sound approach (which polymorph's property is predicted?).
- While I agree the overall crystal contacts/packing dictate mechanical properties (ignoring defects), I do not see sufficient justification as to why predicting unobstructed planes/orthogonal planes/H-bonds bridging is a strong proxy for plasticity of co-crystals, let alone tabletability (there are just so many different types of possible interactions and/or indicators). The citations there are at best weak and do not support the claims.
- The definition of validity/novelty is very relaxed. Validity that is 'this molecule has the right valence' does not rule out thermodynamically infeasible molecules (at least, QED/SAScore should both be reported; otherwise, by definition, you can use SELFIES to get 100% valid molecules with every method). Novelty should not be 'how many molecules are not exactly the same as the training set' but rather a histogram of the closest Tanimoto similarity of the generated molecules to the training set (because a molecule can differ by 1 atom and this metric would still count it as novel). Duplicates are better represented by diversity (histogram of Tanimoto similarity between the generated molecules). Sec 5.4 and Table 4 show very similar/simple aliphatic carboxylic acids. This worries me that the molecules produced lack diversity and are not novel.
- It is very hard to validate the results without wet lab results. The predicted probability of co-crystallization and predicted mechanical properties are unfortunately not verifiable outside of models trained here unless structures are known. While I fully agree that experimental validation should not appear in NeurIPS, I fear that 'generating co-crystals with good tableability' cannot be claimed unless there are wet lab results.
2. The ML approaches taken here are not novel, and to an extent, questionable. If the authors developed a crystallization/property predictor (which produces a combination of scores), I feel the typical optimization approaches (e.g. reinvent, or at least the suite of software in Guacamol) should be at least used as baselines, and otherwise it is hard to justify the usefulness of VAE/GAN; the comparisons against GPT-2 models are much less relevant.

Note: citations on the chemistry side can be significantly improved (e.g. all of the citations 1-6 are hardly relevant, e.g. there are plenty of impactful reviews for charge transfer co-crystals and their applications). I also disagree with the current screening for co-crystals 'focus on rather narrow classes of candidate compounds' (especially when the demonstrated results in the paper are all aliphatic carboxylic acids).

**Questions:**

1. I note that in Sec 5.1, it says the data were randomly split. As we know, co-crystal molecules typically have limited diversity, can the authors elaborate how the predictions generalize out of distribution (e.g. by scaffold splitting)?
2. Can the author produce novelty/diversity of generated samples as distribution of Tanimoto similarities?
3. Could you provide CCDC access code (as is tradition) for Table 3?

**Limitations:**

yes

---

> ### Author Rebuttal · Authors · 2024-08-07
>
> We thank the reviewer for providing a very valuable feedback!
> We believe that some of the criticism was caused by a misunderstanding.
>
> We would like to first __comment on the weaknesses__ outlined by the reviewer:
> 1. Polymorphism is certainly an important factor in the design of co-crystals as it can influence their physicochemical properties [1]. However, the percentage of polymorphs in the CCDC data does not exceed 5% of the total number of co-crystals. Thus, we can assume that their influence on the accuracy of the model in predicting mechanical properties is insignificant.
> 2. In Appendix C.2, we attempted to summarize the relationship of the predicted mechanical properties to the plasticity and tabletability of the co-crystal. Supporting evidence is given by the paper by Bryant et al [2]. This work was published in CrystEngComm, a top-5 domain journal (https://scholar.google.com/citations?view_op=top_venues&hl=en&vq=chm_crystallographystructuralchemistry). So, we respectfully disagree that this is not a sufficient proof of the relationship. Both the publication and the venue are well respected in the crystallographic community.
> 3. We agree with the reviewer's assertion regarding the limitations of relying solely on validity. As suggested by the reviewer, we refined our molecule selection by incorporating the SA score. In Appendix F.5, we mention a threshold of SA ≤ 3, resulting in an average coformer synthesizability value of 2.06.
> 4. The reviewer's concern regarding the high similarity of the molecules in Table 4 (Appendix B.2) is a misunderstanding. In our experiment to discover new coformers for Nicorandil, our objective was to identify coformers with target properties and a high degree of similarity to the existing co-crystals of this drug. Given that the known coformers for Nicorandil are Fumaric and Suberic acid [3], it is logical that the table includes aliphatic carboxylic acids. This result was achieved as intended in the experiment.
> 5. During validation (Section 5.4), we identified three co-crystalline systems that were not included in the training dataset. By showcasing GEMCODE's ability to produce new co-crystal structures reported in the literature as experimentally validated, we prove its predictive capabilities. We believe that our approach to validating GEMCODE is reasonable and convincing. We share the view that wet lab experiments must not be required for NeurIPS application papers. Nevertheless, we currently expand our team to work on experimental validation of our predictions in the lab.
> 6. We are thankful to the reviewer for acknowledging that our work is _"one of the first efforts to utilize generative ML for designing co-formers"_ and addresses _"a very interesting and impactful problem in pharmaceutical production"._ We totally share this assessment. Given that the problem is novel, the applicability of previous approaches is very limited. Existing Guacamol test suites do not support co-crystal design tasks, that is why we did not include the Guacamol benchmarks. The drug design methods implemented in Guacamol (SMILES LSTM, Graph GA) [6] served as an inspiration for our own fine-tuned baselines (we use the GAN-LSTM). REINVENT4 (the latest version) is also not suitable for end-to-end co-crystal design tasks out-of-the-box. The models used in the RL optimizer would have to be retrained and the property prediction part would have to be significantly modified to make a comparison. Additionally, there are works claiming that RL methods are not efficient enough for crystal design [7]. A comprehensive comparison of such methods to ours would certainly be interesting but beyond the scope of this study.
>
> We would like to also provide __answers to the questions__:
> 1. During the validation experiments, GEMCODE successfully generated three new cocrystal systems that were not present in the training data. These cocrystals were reported in the literature as experimentally confirmed. This perfectly demonstrates the generalization capability of our pipeline. Furthermore, in the field of cocrystals, the conventional approach does not typically involve data splits based on class distribution. A recent study accepted for the ICML 2024 workshop [8] employed a standard random split.
> 2. The reviewer's comment regarding the novelty assessment was particularly helpful, so we conducted further analysis and created histograms illustrating the distribution of the maximum Tanimoto Similarity (IT) between the generated coformers and the coformers from the training dataset (https://anonymous.4open.science/r/GEMCODE/rebuttal/histogramms/GAN.png, https://anonymous.4open.science/r/GEMCODE/rebuttal/histogramms/VAE.png, https://anonymous.4open.science/r/GEMCODE/rebuttal/histogramms/CVAE.png). We observed that the distribution is predominantly centered on IT values ranging from 0.5 to 0.6 across all generative models. This observation strongly supports the assertion that the generated molecules exhibit substantial novelty.
> 3. We thank the reviewer for highlighting the importance of adding refcodes to Table 3, as this will enhance the accessibility of the information for the reader. We will update the table in the camera-ready submission.
>
> In conclusion, we would like to highlight that we are positioning GEMCODE as a global open-source project. We value constructive criticism and will use the reviewers’ feedback to improve our methods for data analysis, generative models, and validation of results.
>
> As the reviewer recognized the originality and significance of our work, we kindly ask to consider increasing the score to 6. A NeurIPS publication could attract much attention and interest of the professional community, which is essential for advancing and promoting GEMCODE. With current scores at 8/6/4/4, a moderate increase in the score would significantly increase our chances to get accepted.
>
> __References__ will be posted as a separate official comment due to the limit of rebuttal size.

---

> > ### Comment · Reviewer_XDyo · 2024-08-09
> >
> > I very much appreciate the authors for the comprehensive rebuttal and the additional plots. Unfortunately, my key reservations have not been solved:
> >
> > 1. While the authors suggest that the influence of polymorphism on predicting mechanical properties is insignificant due to its low reporting percentage, I must respectfully disagree. Polymorphism is very much underreported because chemists often do not explore further once a crystal structure is identified (especially in non-pharmaceutical contexts). In crystal structure prediction literature, hundreds of local energy minima are almost always predicted for any given compound. Often the case, these predictions are later validated experimentally.
> > 2. For the correlation between crystal structures' mechanical properties and tableatability, can the authors explain how the diversity of intermolecular interaction affects the prediction? Here, the authors only consider a few criteria, but I am very certain crystals with e.g. halogen bonds would exhibit no hydrogen bonds between planes (hence good score here) but in reality, act very similar to hydrogen bonds. The cited CrystEngComm paper analyzed only 30 crystals and indeed claimed 'While this tool was not intended to be used alone as a predictor of mechanical properties, it _seems to correlate well_ with mechanical properties'. For reasons here and the point above, I do think the claim of tableatability needs to be validated with wet lab results (and hence more suitable for another venue).
> > 3. In Section 5.4, the authors highlight the generation of three new coformers, all of which are carboxylic acids. How is the diversity (in terms of Tanimoto similarity) between generated structures (other than novelty, and I appreciate the histograms)?
> > 4. Maybe this is a misunderstanding, but if the models predict the scores given a pair of SMILES string, why can the authors not employ typical baselines such as Guacamol? Sure, the search space would have to be different and you cannot retrain your scoring function, but GraphGA can surely yield molecules with optimized scores.

---

> > > ### Author Response · Authors · 2024-08-12
> > >
> > > We greatly appreciate the reviewer’s thoughtful comments on our paper.
> > > Below, we provide additional considerations and experimental results to resolve the remaining issues:
> > >
> > > 1. We acknowledge that there were inaccuracies in our response. Undoubtedly, polymorphism of cocrystals is a significant issue. We will include statements emphasizing its importance for pharmaceuticals. However, since our approach is data-driven, and the data on polymorphism is limited, it was impossible to incorporate it into the predictive models in a meaningful way. We will update the Limitations section accordingly. Thanks to the reviewer, we now identify the influence of polymorphism as a prospective research direction.
> > >
> > > 2. We fully agree with the reviewer that, in general, there are a number of crystal parameters to consider. However, since we are looking for co-crystals in the field of pharmaceuticals, directional interactions other than hydrogen bonds (such as halogen bonds, chalcogen bonds, pnictogen bonds) are rare. Considering the full set of such interactions is beyond the scope of the study. Certainly, for better generalization of the approach, the set of descriptors for the interplanar criterion should be further improved, but this requires additional data (i.e., those co-crystals having halogen, chalcogen bonds, etc.), currently not available. Although GEMCODE is a stable solution for pharmaceutical co-crystals, the pipeline may not fit so well for other co-crystals, which we will discuss in the updated Limitations section.
> > >
> > > 3. In response, we plotted distributions of Tanimoto similarity between the molecules generated by the GAN (https://anonymous.4open.science/r/GEMCODE/rebuttal/self_histograms/GAN2.png), and the transformer-based VAE (https://anonymous.4open.science/r/GEMCODE/rebuttal/self_histograms/VAE2.png) and CVAE (https://anonymous.4open.science/r/GEMCODE/rebuttal/self_histograms/CVAE2.png). For each model, the average Tanimoto similarity is between 0.70 and 0.75. On one hand, this highlights sufficient diversity of the molecules. On the other hand, relatively high average similarity was expected, since all generated coformers relate to the same drug. The latter fact aligns well with the observation that the distribution of CVAE is shifted towards 1 due to the “condition” block of the architecture enforcing the target properties of coformers specific to the drug.
> > > We thank the reviewer for suggesting this analysis. We find it very useful, as it supports our key findings but provides additional insights. We acknowledge a moderate fraction of “duplicates” with Tanimoto similarity = 1 among the predicted molecules. Dropping such molecules brings down the percent of target coformers of Table 1 to 2.23%, 1.68% and 5.63% for GAN, T-VAE and T-CVAE, respectively. Please note that these adjustments do not qualitatively change our results and conclusions. We will update the corresponding sections of the paper accordingly.
> > >
> > > 4. We agree that baselines from Guacamol (https://github.com/BenevolentAI/guacamol_baselines) can be used to optimize any molecule in SMILES notation with a given objective. However, unlike the Guacamol tasks, the co-crystal design task is multi-objective. The algorithms from Guacamol_baselines (e.g., the noted GraphGA) are focused on single-objective tasks. There are papers where existing Guacamol tasks are used in the mutli-objective formulation, but they require different optimization techniques [1].
> > > Nevertheless, __we developed a multi-objective modification of GraphGA__ (with Pareto dominance based fitness; the implementation is available at https://anonymous.4open.science/r/GEMCODE/GraphGA_baseline/graphga.py), suitable for co-crystal design tasks. In the __new experiments__, we started from a random subset of co-crystals (with the same population size and number of iterations as was used in GEMOL). GraphGA demonstrated inferior results to GEMOL in terms of target mechanical properties (see plots at https://anonymous.4open.science/r/GEMCODE/GraphGA_baseline/GEMOL_vs_GraphGA.png). More specifically, the highest average probability over all runs was 0.94 (GraphGA) vs 0.95 (GEMOL) for unobstructed planes, 0.63 vs 0.72 for orthogonal planes and 0.12 vs 0.18 for h-bonds bridging. In addition, the convergence of GraphGA was unstable. Ultimately, the hybrid approach of GEMCODE resulted in 21.3% of target molecules on average, while GraphGA produced 20.5% of target molecules of lower quality in terms of predicted mechanical properties. Therefore, we conclude that GEMCODE clearly outperforms the multi-objective version of GraphGA. We will add an appendix section to the camera-ready submission to highlight this comparison. Also, we will further investigate options to extend the set of baselines.
> > >
> > > In light of all this, we kindly ask the reviewer to update the initial review.
> > >
> > > [1] Optimized drug design using multiobjective evolutionary algorithms with SELFIES //arXiv preprint arXiv:2405.00401. - 2024.

---

> ### Author Response · Authors · 2024-08-07
> **References used in the rebuttal**
>
> __References:__
>
> [1] Heng, T., Yang, D., Wang, R., Zhang, L., Lu, Y., & Du, G. (2021). Progress in research on Artificial Intelligence applied to polymorphism and cocrystal prediction. ACS omega, 6(24), 15543-15550.
>
> [2] Bryant, M. J., Maloney, A. G. P., & Sykes, R. A. (2018). Predicting mechanical properties of crystalline materials through topological analysis. CrystEngComm, 20(19), 2698-2704.
>
> [3] Mannava, M. C., Gunnam, A., Lodagekar, A., Shastri, N. R., Nangia, A. K., & Solomon, K. A. (2021). Enhanced solubility, permeability, and tabletability of nicorandil by salt and cocrystal formation. CrystEngComm, 23(1), 227-237.
>
> [4] Tripp, Austin, and José Miguel Hernández-Lobato. "Genetic algorithms are strong baselines for molecule generation." arXiv preprint arXiv:2310.09267 (2023).
>
> [5] Ye Z. H. et al. Searching new cocrystal structures of CL-20 and HMX via evolutionary algorithm and machine learning potential //Journal of Materials Informatics. – 2024. – Т. 4. – №. 2.
>
> [6] Brown N. et al. GuacaMol: benchmarking models for de novo molecular design //Journal of chemical information and modeling. – 2019. – Т. 59. – №. 3. – С. 1096-1108.
>
> [7] Thomas M. et al. Augmented Hill-Climb increases reinforcement learning efficiency for language-based de novo molecule generation //Journal of cheminformatics. – 2022. – Т. 14. – №. 1. – С. 68.
>
> [8] Birolo, R., Özçelik, R., Aramini, A., Gobetto, R., Chierotti, M. R., & Grisoni, F. (2024). Deep Supramolecular Language Processing for Co-crystal Prediction. (https://openreview.net/forum?id=bQ9d2hzjW4&noteId=9SoErgR0kb)

---

### Author Response · Authors · 2024-08-14

Dear Reviewers and Area Chairs,

As the end of the discussion period is approaching, we would like to say a few words.

We extend our sincere gratitude to the reviewers. __Thank you very much for your work!__ Your insightful comments and constructive criticism prompted us to significantly improve the quality of our paper. As mentioned before, we will provide additional clarifications, discussions, as well as new experimental results supporting our original claims in the camera-ready submission.

We are very happy to have addressed most (if not all) of the concerns raised in the initial reviews, which resulted in __3/4 reviewers recommending to accept our work__ for publication with an __average rating of 6.0__. We would like to thank the reviewers also for actively engaging in the discussions!

Notably, regardless of the initial review ratings, __all the reviewers acknowledged the novelty and significance of our work__. We are very pleased to have received such evaluations and kindly ask the area chairs and meta-reviewers to take this into account while making the final decision on the paper. A few quotes highlighting this point are:

  - _"The manuscript tackles a very interesting and impactful problem in pharmaceutical production, and overall a long-standing challenge in crystal engineering. To my knowledge, this is one of the first efforts to use generative ML to design co-formers..."_
  - _"...this is a novel task that the authors have decided to tackle with GEMCODE."_
  - _"This paper highlights an important application of generative AI to a new domain."_
  - _"This paper is the first to introduce generative AI into co-crystal design, which is a very important topic for drug development and other fields."_
  - _"...this paper is a pioneer in both the property prediction and generation/optimization of co-crystals..."_

We keep working on the refinements to GEMCODE and look forward to the author notification!

Best regards,
Authors

---

### Decision · Program_Chairs · 2024-09-25

**Decision:**

Accept (poster)

**Comment:**

This paper proposes GEMCODE: a pipeline to design novel coformers that could form tabletable co-crystals with a given organic molecule. GEMCODE is comprised of a sequence of approaches, including generative and predictive models, as well as evolutionary algorithms. The authors conduct an empirical coformer design study using known drugs, showing that GEMCODE can both rediscover known coformers and also propose promising new ones.

The paper sparked significant debate among the reviewers, both during the author rebuttal period, as well as later during the inter-reviewer discussion. Notably, one reviewer consistently championed the work (rating of “Strong Accept”), and one leaned slightly towards rejection; the other two reviewers were undecided (shifting their ratings up during rebuttal but then down during reviewer discussion) but ended up leaning slightly towards acceptance.

Main positive points raised by the reviewers were as follows:

-	This paper is one of the first efforts in its domain (subdomain of the broader “AI for Science”), and likely would be the first one accepted at a conference such as NeurIPS. Its acceptance could generate more interest in applying ML to an impactful scientific problem.
-	The paper is well-written, thorough, and makes extensive use of existing approaches/datasets where appropriate (it does not strive for novelty for novelty’s sake).
-	Experimental results, albeit purely in silico, appear promising.

Some of the issues highlighted by reviewers were addressed by authors during the rebuttal, and hence are not further discussed here. Main complaints that were not resolved are as follows (each accompanied by a counterpoint raised in discussion):

-	To confirm GEMCODE is fully practical, it would be great for the work to (a) predict/consume crystal structure instead of SMILES alone to account for polymorphism, and (b) include wetlab experiments to verify the results. While everyone generally agreed this would be ideal, the positive reviewer argued the work is already useful as a starting point for future research. However, reviewers noted that, in the presence of these limitations, the claims made in this work may appear overconfident.
-	ML approaches used in this paper are generally not novel (mostly simple adaptations of existing ideas), and the work appears more valuable from the domain side, raising questions as to whether it’s appropriate for NeurIPS. While I agree with the lack of novelty, my own comment on this would be to keep in mind that “AI for Science” is a strong theme at NeurIPS-like conferences, and while most accepted papers in this domain do have significantly more novelty than GEMCODE, they unfortunately are often of limited practical use, precisely because NeurIPS sometimes tends to select for novelty over practicality. If anything, accepting a more practical and less methodologically novel work could correct for this bias. Future works on co-crystal design would have something to compare to, and likely also bring more ML novelty.

Summing up, this paper generated a good deal of healthy disagreement between the reviewers. I think the arguments from both sides are perfectly reasonable and could see this paper falling on either side of the acceptance bar. **As the paper pioneers a new subdomain of “AI for Science” that could be of interest to the NeurIPS community, and it does so in a practical way, I am willing to lean into its positive aspects, and recommend the acceptance of the paper.**
That being said, the complaints listed above remain, and I would **urge the authors to make a serious attempt at integrating all the review feedback** into their work. In particular, please make the limitations of GEMCODE (e.g. not dealing with polymorphism, lack of experimental validation, etc) clearer, and discuss the extra references and points of view provided by the reviewers.